

# Evaluation of simulated biomass damage in forest ecosystems induced by ozone against observation-based estimates

Martina Franz[1,2], Rocio Alonso[4], Almut Arneth[5], Patrick Büker[6], Susana Elvira[4], Giacomo Gerosa[7], Lisa Emberson[6], Zhaozhong Feng[8], Didier Le Thiec[9], Riccardo Marzuoli[7], Elina Oksanen[10], Johan Uddling[11], Matthew Wilkinson[12], and Sönke Zaehle[1,3]

[1]Biogeochemical Integration Department, Max Planck Institute for Biogeochemistry, Jena, Germany
[2]International Max Planck Research School (IMPRS) for Global Biogeochemical Cycles, Jena, Germany
[3]Michael Stifel Center Jena for Data-driven and Simulation Science, Jena, Germany
[4]Ecotoxicology of Air Pollution, CIEMAT - Research Center for Energy, Environment and Technology, Avda. Complutense 40, edif.70, Madrid 28040, Spain
[5]Karlsruhe Institute of Technology (KIT), Department of Atmospheric Environmental Research (IMK-IFU), Garmisch-Partenkirchen, Germany
[6]Stockholm Environment Institute at York, Environment Dept., University of York, YO10 5NG, United Kingdom
[7]Department of Mathematics and Physics, Catholic University of Brescia, via Musei 41, Brescia (Italy)
[8]State Key Laboratory of Urban and Regional Ecology, Research Center for Eco-Environmental Sciences, Chinese Academy of Sciences, Shuangqing Road 18, Haidian District, Beijing, 100085, China
[9]Inra, Université de Lorraine, AgroParisTech, Silva, F-54280 Champenoux, France
[10]Department of Environmental and Biological Sciences, University of Eastern Finland, 80101 Joensuu, Finland
[11]Department of Biological and Environmental Sciences, University of Gothenburg, Gothenburg, Sweden
[12]Centre for Sustainable Forestry and Climate Change, Forest Research, UK

**Correspondence:** Martina Franz (mfranz@bgc-jena.mpg.de)

**Abstract.**

Regional estimates of the effects of ozone pollution on forest growth depend on the availability of reliable damage functions that estimate a representative ecosystem response to ozone exposure. A number of such damage functions for forest tree species and forest functional types have recently been published and subsequently applied in terrestrial biosphere models to

estimate regional or global effects of ozone on forest tree productivity and carbon storage in the living plant biomass. The resulting impacts estimated by these biosphere models show large uncertainty in the magnitude of ozone effects predicted. To understand the role that these damage functions play in determining the variability of estimated ozone impacts, we use the O-CN biosphere model to provide a standardised modelling framework. We test four published damage functions describing the leaf-level, photosynthetic response to ozone exposure (targeting $V_{cmax}$ or net photosynthesis) in terms of their simulated

whole-tree biomass responses against field data from 23 ozone filtration/fumigation experiments conducted with European tree species at sites across Europe with a range of climatic conditions. Our results show that none of these previously published damage functions lead to simulated whole-tree biomass reductions in agreement with the observed dose-response relationships derived from these field experiments, and instead lead to significant over- / or underestimations of the ozone effect. By re-parameterising these photosynthetic based damage functions we develop linear, plant functional type specific dose-response

relationships, which provide accurate simulations of the observed whole-tree biomass response across these 23 experiments.





# 1 Introduction

Ozone is a phytotoxic air pollutant which enters plants mainly through the leaf stomata, where reactive oxygen species (ROS) are formed that can damage essential leaf functioning (Ainsworth et al., 2012). Ozone induced declines in net photosynthesis (Morgan et al., 2003; Wittig et al., 2007) have been observed as the result of damage of the photosynthetic apparatus, increased

respiration rates caused by investments in repair of injury, as well as the production of defence compounds (Wieser and Matyssek, 2007; Ainsworth et al., 2012). At the leaf-scale, ozone damage occurs and accumulates, when the instantaneous stomatal ozone uptake of leaves surpasses the ability of the leaf to detoxify ozone (Wieser and Matyssek, 2007). These effects are likely the primary cause for reduced rates of net photosynthesis and decreased supply of carbon and energy for growth and net primary production (NPP), which contributes to the commonly observed ozone-induced reductions in leaf area and plant

biomass (Morgan et al., 2003; Lombardozzi et al., 2013; Wittig et al., 2009). Changes in tropospheric ozone abundance and associated changes in ozone-induced damage thus have the potential to affect the ability of the terrestrial biosphere to sequester carbon (Harmens and Mills, 2012; Oliver et al., 2017). However, a quantitative understanding of the effect of ozone pollution on forest growth and carbon sequestration at the regional scale is still lacking. Terrestrial biosphere models can be used to obtain regional or global estimates of ozone damage based on an understanding of how ozone affects plant processes leading to

C assimilation and growth. Modelling algorithms to estimate regional or global impacts of ozone on gross primary production (GPP) have been developed for several of these terrestrial biosphere models (Sitch et al., 2007; Lombardozzi et al., 2012a, 2015; Franz et al., 2017; Oliver et al., 2017). However, simulated reductions in GPP due to ozone damage vary substantially between models and model versions.

This uncertainty is predominantly due to the different approaches that these models use to relate ozone uptake (or ozone

exposure) to reductions in whole-tree biomass, and in the exact parameterisation of the dose-response relationship applied (Karlsson et al., 2004; Pleijel et al., 2004; Wittig et al., 2007; Lombardozzi et al., 2012a, 2013). The dose-response relationships employed by current terrestrial biosphere models differ decidedly in their slope (i.e. the change in damage per unit of time-integrated ozone uptake), intercept (ozone damage at zero time-integrated ozone uptake), and in their assumed threshold, below which the ozone uptake rate is considered sufficiently low that ozone will be detoxified before any damage occurs (Karlsson

et al., 2004; Pleijel et al., 2004; Lombardozzi et al., 2012a). For example, Sitch et al. (2007) relates the instantaneous ozone uptake exceeding a flux threshold to net photosynthetic damage via an empirically derived factor. An alternative approach has been to relate ozone damage to net photosynthesis in response to the accumulated ozone uptake rather than to the instantaneous ozone uptake as in Sitch et al. (2007), e.g. by using the $CUOY$, which refers to the cumulative canopy $O_3$ uptake above a flux threshold of $Y\ nmol\,m^{-2}\,s^{-1}$ (Wittig et al., 2007; Lombardozzi et al., 2012a, 2013).

The effect of ozone on plant growth has been investigated by ozone filtration/fumigation experiments either at the individual experimental level or by pooling data from multiple experiments that have been conducted according to standardised experimental method. These experiments typically rely on small trees or saplings. A challenge in developing and testing process-based models of ozone damage from these ozone fumigation experiments is that often only the difference in biomass accumulation between plants grown in an ozone treatment and in ambient or charcoal-filtered air at the end of the experiment





are reported. Data from these studies provide evidence for a linear, species-specific relationship between accumulated ozone uptake and reductions in plant biomass (Pleijel et al., 2004; Mills et al., 2011; Nunn et al., 2006, e.g.). Sitch et al. (2007) for instance calibrated their instantaneous leaf-level dose-response relationship between ozone uptake and photosynthesis by relating simulated annual net primary production and accumulated ozone uptake to observed biomass dose-response relation-

ships developed by Karlsson et al. (2004) and Pleijel et al. (2004), where biomass/yield damage is related to the Phytotoxic Ozone Dose ($PODy$). The $PODy$ refers to the accumulated ozone uptake above a flux threshold of $y\,nmol\,m^{-2}\,s^{-1}$ by the leaves representative of the upper canopy leaves of the plant. Such an approach applies biomass dose-response relationships of young trees to mature trees. However, the effects of ozone on leaf physiology (e.g. net photosynthesis and stomatal conductance) or plant carbon allocation may differ between juvenile and adult trees (Hanson et al., 1994; Samuelson and Kelly, 1996;

Kolb and Matyssek, 2001; Paoletti et al., 2010). Whether or not biomass dose-response relationships can be used to calibrate dose-response functions for mature trees is uncertain.

An alternative approach is to directly simulate ozone damage to photosynthesis, which may have been a major cause for the observed decline in plant biomass production (Ainsworth et al., 2012). Possible damage targets in the simulations can be for example the net photosynthesis or leaf-specific photosynthetic activity (such as represented by the maximum carboxylation

capacity of Rubisco, $V_{cmax}$). For instance Lombardozzi et al. (2012a) based their dose-response relationships on an experimental study involving a single forest tree species, whereas more recent publications (e.g. Lombardozzi et al. (2015) and Franz et al. (2017)) have used dose-response relationships from meta-analyses of a far larger-set of filtration/fumigation studies. Meta-analyses have attempted to summarise the responses of plant performance to ozone exposure across a wider range of experiments and vegetation types (Wittig et al., 2007; Lombardozzi et al., 2013; Feng and Kobayashi, 2009; Li et al., 2017;

Wittig et al., 2009) and to develop damage functions for plant groups that might provide an estimate of mean plant group responses to ozone. However, these meta-analyses suffer from a lack of consistency in the derivation of either plant damage or ozone exposure, and generally report a large amount of unexplained variance. A further complication in the meta-analyses of ozone damage (e.g. Wittig et al., 2007; Lombardozzi et al., 2013) is that they have to indirectly estimate the cumulative ozone uptake underlying the observed ozone damage based on a restricted amount of data, which causes uncertainty in the derived

damage functions.

Büker et al. (2015) provides an independent data set of whole-tree biomass plant responses to ozone uptake which is independent of data sets that were used to describe damage functions by Wittig et al. (2007) and Lombardozzi et al. (2013). This data set has been collected from experiments that follow a more standardised methodology to assess dose-responses and has associated meteorological and ozone data at a high time resolution that allow more accurate estimates of modelled ozone

uptake to be made. These dose-response relationships describe whole-tree biomass reductions in tree seedlings derived from standardised ozone filtration/fumigation methods for eight European tree species at ten locations across Europe (see Tab. A.2 for details Büker et al., 2015). These data thus provide an opportunity to evaluate simulations of biosphere models that use leaf level damage functions (describing the effect of ozone uptake on photosynthetic variables) to estimate C assimilation, growth and ultimately whole tree biomass against these robust empirical dose-response relationships that relate ozone exposure directly

to whole tree biomass response.





Here we test four alternative, previously published ozone damage functions that target either net photosynthesis or the leaf carboxylation capacity ($V_{cmax}$), which have been included in state-of-the-art terrestrial biosphere models (Lombardozzi et al., 2012a, 2015; Franz et al., 2017) against these new biomass dose-response relationships by Büker et al. (2015). We incorporate these damage functions into a single modelling framework, the O-CN model (Zaehle and Friend, 2010; Franz et al., 2017).

To reduce model-data mismatch, we test the functions in simulations that mimic to the extend possible the conditions of each of the experiments in the Büker et al. (2015) data-set, in particular the young age, such that we can directly compare the simulated to the observed whole-tree biomass reductions of the empirically derived dose-response relationships. This allows us to identify the contribution of these alternative damage function formulations on the simulated whole-tree biomass response. The simulated biomass dose-response relationships are then compared to the data from the experiments to evaluate

the capability of the different model versions to reproduce observed dose-response relationships. Based on these comparisons we use a similar approach to that of Sitch et al. (2007) and develop alternative parameterisations of the damage functions to improve the capability of the O-CN model to simulate the whole-tree biomass responses observed in the field experiments, with the notable exception that we explicitly simulate in-fumigation experiments and the approximate age of the trees. Finally, we explore whether or not there is a substantial difference in the biomass response to ozone of young or mature trees by using

a sequence of model simulations and comparing the response both in terms of whole tree biomass as well as net primary production.

## 2   Methods

We use the O-CN terrestrial biosphere model (Zaehle and Friend, 2010), which is an extension of the ORCHIDEE model (Krinner et al., 2005) to simulate conditions of the ozone fumigation experiments described in Büker et al. (2015). The O-CN

model simulates the terrestrial coupled carbon (C), nitrogen (N) and water cycles for up to twelve plant functional types and is driven by climate data and atmospheric composition.

O-CN simulates a multi-layer canopy with up to 20 layers with a thickness of up to 0.5 leaf area index each. Net photosynthesis is calculated according to a modified Farquhar-scheme for shaded and sun-lit leaves considering the light profiles of diffuse and direct radiation (Zaehle and Friend, 2010). Leaf nitrogen concentration and leaf area determine the photosynthetic

capacity. Increases of the leaf nitrogen content increase $V_{cmax}$ and $J_{max}$ (nitrogen specific rates of maximum light harvesting, electron transport) and hence maximum net photosynthesis and stomatal conductance per leaf area. The leaf N content is highest at the top of the canopy and exponentially decreases with increasing canopy depth. Following this net photosynthesis, stomatal conductance and ozone uptake are generally highest in the top canopy and decrease with increasing canopy depth.

Canopy-integrated assimilated carbon enters a labile non-structural carbon pool, which can either be used to fuel mainte-

nance respiration (a function of tissue nitrogen), storage (for seasonal leaf and fine root replacement and buffer of inter-annual variability of assimilation) or biomass growth. After accounting for reproductive production (flowers and fruits), biomass growth is partitioned into leaves, fine roots, and sapwood according to a modified pipe-model (Zaehle and Friend, 2010), accounting for the costs of biomass formation (growth respiration). In other words, changes in leaf-level productivity affect the



build-up of plant pools and storage, and thereby feed back on the ability of plants to acquire C through photosynthesis, or nutrients through fine root uptake.

## 2.1 Ozone damage calculation in O-CN

Leaf-level ozone uptake is determined by stomatal conductance and atmospheric $O_3$ concentrations, as described in Franz et al. (2017). To mimic the conditions of the fumigation experiments with plot-level controlled atmospheric $O_3$ concentrations, simulations are conducted with a model version of O-CN, in which atmospheric $O_3$ concentrations are directly used to calculate ozone uptake into the leaves, and the transfer and destruction of ozone between the atmosphere and the surface is ignored (ATM model version in Franz et al. (2017)). Deviating from Franz et al. (2017), stomatal conductance $g_{st}$ here is calculated based on the Ball and Berry formulation (Ball et al., 1987) as

$$g_{st,l} = g_0 + g_1 \times \frac{A_{n,l} \times RH \times f(height_l)}{C_a} \tag{1}$$

where net photosynthesis ($A_{n,l}$) is calculated as described in Zaehle and Friend (2010) as a function of leaf nitrogen and nitrogen specific rates of maximum light harvesting, electron transport ($J_{max}$) and carboxylation rates ($V_{cmax}$). $RH$ is the atmospheric relative humidity, $f(height_l)$ the water-transport limitation with canopy height, $C_a$ the atmospheric $CO_2$ concentration, $g_0$ is the residual conductance when $A_n$ approaches zero, and $g_1$ is the stomatal-slope parameter as in Krinner et al. (2005). The index $l$ indicates that $g_{st}$ is calculated separately for each canopy layer.

The $O_3$ stomatal flux ($f_{st,l}$, $nmol\,m^{-2}(leaf\,area)\,s^{-1}$) is calculated from the atmospheric $O_3$ concentration the plants in the field experiments were fumigated with ($\chi_{atm}^{O_3}$) and $g_{st,l}$ as

$$f_{st,l} = (\chi_{atm}^{O_3} - \chi_i^{O_3})g_{st,l}^{O_3}. \tag{2}$$

where the leaf internal $O_3$ concentration ($\chi_i^{O_3}$) is assumed to be zero (Laisk et al., 1989).

The accumulation of ozone fluxes above a threshold of $Y$ $nmol\,m^{-2}(leaf\,area)\,s^{-1}$ ($f_{st,l,Y}$, $nmol\,m^{-2}(leaf\,area)\,s^{-1}$) with

$$f_{st,l,Y} = MAX(0, f_{st,l} - Y) \tag{3}$$

gives the $CUOY_l$. The canopy value of $CUOY$ is calculated by summing $CUOY_l$ over all canopy layers (see Franz et al. (2017) for details).

For comparison to observations, the Phytotoxic Ozone Dose ($POD$, $mmol\,m^{-2}$) can be diagnosed by the accumulation of $f_{st,l}$ for the top canopy layer ($l = 1$). The accumulation of ozone fluxes of the top canopy layer above a threshold of $y$ $nmol\,m^{-2}(leaf\,area)\,s^{-1}$ gives the $PODy$. The estimates of $PODy$ (both $POD2$ and $POD3$) can be used off-line to re-





construct dose-response relationships equivalent to those described in Büker et al. (2015). These modelled dose-response relationships can then be compared with the empirically derived dose-response relationships to assess the ability of the model to estimate damage. As such, the $POD2$ and $POD3$ used for the formation of these modelled dose-response relationships are purely diagnostic variables and not involved in the damage calculation of the model. The flux thresholds (2 and 3 $nmol\,m^{-2}(leaf\,area)\,s^{-1}$) are not the flux thresholds that are used to estimate biomass response in the O-CN model simulations.

Ozone damage, i.e. the fractional loss of carbon uptake associated with ozone uptake $d_l^{O_3}$, is calculated as a linear function of the cumulative leaf-level uptake of ozone above a threshold of $Y\,nmol\,m^{-2}(leaf\,area)\,s^{-1}$ ($CUOY_l$)

$$d_l^{O_3} = a - b \times CUOY_l \qquad (4)$$

where $a$ is the intercept and $b$ is the slope of the damage function. The damage fraction ($d_l^{O_3}$) is calculated separately for each canopy layer $l$ based on the specific accumulated ozone uptake of the respective canopy layer ($CUOY_l$), and takes values between 0 and 1. The magnitude of $d_l^{O_3}$ in Eq. 4 varies between the canopy layers because $CUOY_l$ varies driven by within-canopy gradients in stomatal conductance and photosynthetic capacity.

The effect of ozone damage on plant carbon uptake is calculated by

$$x_{n,l}^{O_3} = x_{n,l}(1 - d_l^{O_3}). \qquad (5)$$

where $x$ is either leaf-level net photosynthesis $A_{n,l}$ or the maximum photosynthetic capacity ($J_{max,l}$ and $V_{cmax,l}$), which is used in the calculation of $A_{n,l}$. $J_{max,l}$ and $V_{cmax,l}$ are reduced in proportion such that the ratio between the two is not altered. While there is some evidence that ozone can affect the ration between $J_{max}$ and $V_{cmax}$, we believe that for the purpose of this paper is is justifiable to assume a fixed ratio between them.

Reductions in $A_n, l$ cause a decline in stomatal conductance ($g_{st,l}$) due to the tight coupling between both. Other stress factors that impact $g_{st,l}$ are accounted for in the preceding calculation of the $g_{st,l}$ undamaged by ozone (see Eq. 1). Reductions in $g_{st,l}$ decrease the $O_3$ uptake into the plant ($f_{st,l}$) and slow the increase in $CUOY_l$ and thus ozone damage.

## 2.2  Model set-up

Four published damage functions were applied within the O-CN model (see Tab. 1 for the respective slopes, intercepts and flux thresholds). As these did not match well with the observed biomass dose-response relationships by Büker et al. (2015), we calibrated two additional damage relationships, one each for $A_n$ or $V_{cmax}$, based on the data presented in Büker et al. (2015) (see Tab. 1 for slopes and intercepts). For these calibrated damage functions, we chose a flux threshold value of 1 $nmol\,m^{-2}(leaf\,area)\,s^{-1}$, as suggested by LRTAP-Convention (2017). We forced the intercept ($a$) of these relationships to one to simulate zero ozone damage at zero accumulated $O_3$ (for ozone levels that cause less then 1 $nmol\,m^{-2}(leaf\,area)\,s^{-1}$



instantaneous ozone uptake). As described above, in all model versions, ozone damage is calculated independently for each canopy layer based on the accumulated $O_3$ uptake ($CUOY_l$) in that layer, above a specific flux threshold of $Y\ nmol\ m^{-2}\ (leaf\ area)\ s^{-1}$ for the respective damage function (see Tab. 1).

**Table 1.** Slopes and intercepts, partly PFT specific, of all four published (W07$_{PS}$, L12$_{PS}$, L12$_{VC}$, L13$_{PS}$) and two tuned (tun$_{PS}$, tun$_{VC}$) damage functions included in O-CN. Targets of ozone damage are net photosynthesis (PS) or $V_{cmax}$. Damage calculations base on the $CUOY$ with a specific flux threshold for each damage function.

| ID | Target | Slope (b) | Intercept (a) | Plant group | Flux threshold $[nmol\ m^{-2}\ (leaf\ area)\ s^{-1}]$ | Reference |
|---|---|---|---|---|---|---|
| W07$_{PS}$ | PS | 0.0022 | 0.9384 | All | 0 | Wittig et al. (2007) |
| L12$_{PS}$ | PS | 0.2399 | 1.0421 | All | 0.8 | Lombardozzi et al. (2012a) |
| L12$_{VC}$ | $V_{cmax}$ | 0.1976 | 0.9888 | All | 0.8 | Lombardozzi et al. (2012a) |
| L13$_{PS}$ | PS | 0 | 0.8752 | Broadleaf | 0.8 | Lombardozzi et al. (2013) |
| L13$_{PS}$ | PS | 0 | 0.839 | Needleleaf | 0.8 | Lombardozzi et al. (2013) |
| tun$_{PS}$ | PS | 0.065 | 1 | Broadleaf | 1 | tuned here |
| tun$_{PS}$ | PS | 0.021 | 1 | Needleleaf | 1 | tuned here |
| tun$_{VC}$ | $V_{cmax}$ | 0.075 | 1 | Broadleaf | 1 | tuned here |
| tun$_{VC}$ | $V_{cmax}$ | 0.025 | 1 | Needleleaf | 1 | tuned here |

## 2.3 Model and protocol

Simulations were run for each fumigation experiment using meteorological input from the daily CRU-NCEP climate data set (CRU-NCEP version 5; LSCE (http://dods.extra.cea.fr/store/p529viov/cruncep/V5_1901_2013/) at the nearest grid cell to the coordinates of the experiment sites. The meteorological data provided by the experiments were incompletely describing the atmospheric boundary conditions required to drive the O-CN model. Atmospheric $CO_2$ concentrations were taken from Sitch et al. (2015), and reduced as well as oxidised nitrogen deposition in wet and dry forms were provided by the EMEP model (Simpson et al., 2014). Hourly $O_3$ concentrations were obtained from the experiments, as in Büker et al. (2015).

Büker et al. (2015) report data for eight tree species at 11 sites across Europe (see Tab. A.2 for experiment and simulation details). The O-CN model simulates twelve plant functional types (PFT's) rather than explicit species, therefore the species from the experiments were assigned to the corresponding PFT: All broadleaved species except *Quercus ilex* were assigned to the temperate broadleaved summergreen PFT. *Quercus ilex* was classified as temperate broadleaved evergreen PFT. All needle-leaved species were assigned to the temperate needle-leaved evergreen PFT.

The field experiments were conducted on young trees or cuttings. Prior to the simulation of the experiment, the model was run in an initialisation phase from bare ground until the simulated stand-scale tree age was stable and representative of 1-2 year old seedlings. During this initialisation, O-CN was run with the climate of the years preceding the experiment and zero atmospheric $O_3$ concentrations. Using ambient ozone concentrations during the initialisation phase would have resulted





in different initial biomass values for the different response functions, which would have reduced the comparability of the different model runs. The impact of the ozone concentrations in the initialisation phase on our results here can be considered negligible since we only evaluate the simulated biomass from different treatments in relation to each other and do not evaluate it in absolute terms.

5    The duration of the initialisation phase depends on the site and PFT and averages 7.8 years (mean over all simulated experiments). Some of the published damage functions and/or parameterisations applied have intercepts unequal to one ($a$ in Eq. 4; see Tab. 1), which induces reductions ($a < 1$) or increases ($a > 1$) in photosynthesis at zero ozone concentration and thus causes a bias in biomass and in particular foliage area at the end of the initialisation phase. To eliminate this bias, the nitrogen-specific photosynthetic capacity of a leaf was adjusted for each of the six parameterisations of the model to obtain comparable LAI values at the beginning of the experiment (see Tab. A.1). This adaption of the nitrogen-specific photosynthetic capacity of a leaf only counterbalances the fixed increases or decreases in the calculation of photosynthesis implied by the intercepts unequal to 1 and has no further impact on ozone uptake and damage calculations.

The simulations of the experiments relied on the meteorological and atmospheric forcing of the experiment years. Simulations were made for all reported $O_3$ treatments of the specific experiment, including the respective control treatments. Büker et al. (2015) obtained estimates of biomass reductions due to ozone by calculating the hypothetical biomass at zero ozone uptake for all experiments that reported ozone concentrations greater than zero for the control group (e.g. for charcoal filtered or non-filtered air) and calculated the biomass damage from the treatments against a completely undamaged biomass. Our model allows us to run simulations with zero ozone concentrations and skip the calculation of the hypothetical biomass at zero ozone concentrations as done by Büker et al. (2015). Following this, we ran additional reference simulations with zero $O_3$ and based our biomass damage calculations upon them.

To test whether biomass dose-response relationships of mature forests will show a similar relationship as observed in the simulations of young trees, we ran additional simulations with mature trees. To allow the development of a mature forest where biomass accumulation reached a maximum, and high, and medium turnover soil pools reached an equilibrium, the model was run for 300 years in the initialisation phase. The simulations were conducted with the respective climate previous to the experiment period and zero atmospheric $O_3$ concentration. For the simulation years previous to 1901 the yearly climate is randomly chosen from the years 1901-1930. Constant values of atmospheric $CO_2$ concentrations are used in simulated years previous to 1750 followed by increasing concentrations up to the experiment years. The subsequent experiment years are simulated identical as in the simulations with the young trees.

## 2.4  Calculation of the biomass damage relationships

30  At each experiment site and for all treatments the annual reduction in biomass due to ozone ($RB$) is calculated as in Büker et al. (2015):

$$RB = \left( \frac{BM_{treat}}{BM_{zero}} \right)^{\frac{1}{n}},$$
(6)





where $BM_{treat}$ represents the biomass of a simulation, which experienced an $O_3$ treatment and $BM_{zero}$ the biomass of the control simulation with zero atmospheric $O_3$ concentration. The exponent imposes an equal fractional biomass reduction across all simulation years for experiments lasting longer than one year.

Büker et al. (2015) report the dose-response relationships for biomass reduction with reference to the Phytotoxic Ozone Dose
($PODy$) with flux thresholds $y$ of 2 and 3 $nmol\,m^{-2}(leaf\,area)\,s^{-1}$ ($POD_2$ and $POD_3$) for the needleleaf and broadleaf category respectively. To be able to compare the simulated biomass reduction by O-CN with these estimates, we diagnosed these $PODy$ values for each simulation from the accumulated ozone uptake of the top canopy layer ($PODy_{O-CN} = CUOY_{l=1}$). Note that the $PODy_{O-CN}$ is purely diagnostic, and not used in the damage calculations, which are based on the $CUOY_l$ (see Eq. 4). As O-CN computes continuous, half-hourly values of ozone uptake (see Franz et al. (2017) for details), the
$PODy_{O-CN}$ values have to be transformed to be comparable to the mean annual $PODy$ values reported in Büker et al. (2015). For deciduous species, the yearly maximum of $PODy_{O-CN}$ was taken as yearly increment $PODy_{O-CN,i}$. The $PODy_{O-CN}$ of evergreen species was continuously accumulated over several years. To obtain the yearly increment $PODy_{O-CN,i}$, the $PODy_{O-CN}$ at the beginning of the year $i$ is subtracted from the $PODy_{O-CN}$ at the end of the year $i$.

The selected yearly $PODy_{O-CN,i}$ were used to calculate mean annual values necessary for the formation of the dose-
response relationships integrating all simulation years ($PODy^{dr}$) as

$$PODy_i^{dr} = \frac{\sum_{k=1}^{i} PODy_{O-CN,i}}{i} \tag{7}$$

where $PODy_{O-CN,i}$ is the $PODy$ of the $i$-th year calculated by O-CN. The $PODy^{dr}$ values are used to derive biomass dose-response relationships.

Separate biomass dose-response relationships were estimated by grouping site data for broadleaved and needleleaved species.
The biomass dose-response relationships are obtained from the simulation output by fitting a linear model to the simulated values of $RB$ and $PODy^{dr}$ (with flux thresholds of 2 and 3 $nmol\,m^{-2}(leaf\,area)\,s^{-1}$ for needleleaved and broadleaved species, respectively), where the regression line is forced through one at zero $PODy^{dr}$. Büker et al. (2015) report two alternative dose-response relationships for their data set, the simple and the standard model, $B_{SI}$ and $B_ST$ respectively. We evaluate our different model versions regarding their ability to reach, with the biomass-dose-response relationships computed from their output, the area between those two functions (target area). The tuned damage relationships $tun_{PS}$ and $tun_{VC}$ were obtained by adjusting the slope $b$ in Eq. 4 such that the corresponding biomass dose-response relationships fits the target area. The intercept of the damage relationships are forced to 1 to simulate zero ozone damage at ozone fluxes lower than 1 $nmol\,m^{-2}(leaf\,area)\,s^{-1}$.





# 3 Results

## 3.1 Testing published damage functions

None of the versions where ozone damage is calculated based on previously published damage functions fit the observations well. Some versions strongly overestimate the simulated biomass dose-response relationship and others strongly underestimate

it (see Fig. 1) compared to the dose-response relationships developed by Büker et al. (2015).

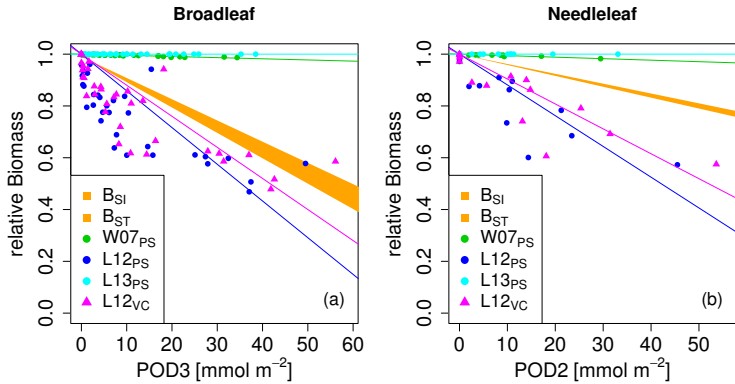

**Figure 1.** Biomass dose-response relationships for simulations based on published damage relationships, separate for a) broadleaved species, and b) needle-leaved species. The dose-response relationships by Büker et al. (2015), $B_{SI}$ and $B_{ST}$, define the target area (orange). The displayed dose-response relationships are simulated by model versions which base damage calculations either on net photosynthesis W07$_{PS}$ (Wittig et al., 2007), L12$_{PS}$(Lombardozzi et al., 2012a), and L13$_{PS}$ (Lombardozzi et al., 2013), or on $V_{cmax}$ L12$_{VC}$ (Lombardozzi et al., 2012a) (see Tab. 1 for more details). See Tab. A.3 and A.4 for slopes, intercepts, $R^2$ and p-values of the displayed regression lines. Damage calculation in the simulations bases on $CUOY$ (see Tab. 1) and not on $POD2$ or $POD3$ (see Sec. 2.4 for more details).

In the W07$_{PS}$ simulations, where damage is calculated based on the damage function by Wittig et al. (2007), biomass damage is strongly underestimated compared to the estimates from Büker et al. (2015). Ozone damage estimates are mainly driven by the intercept of the relationship, which assumes a reduction of net photosynthesis by 6.16% at zero ozone uptake. Little additional ozone damage occurs due to the accumulation of ozone uptake. As a consequence, the ozone treatments and reference simulations differ little in their simulated biomass. Similarly, the Lombardozzi et al. (2013) damage function (L13$_{PS}$)

calculates ozone damage as a fixed reduction of net photosynthesis independent of the actual accumulated ozone uptake. The reference simulations with zero atmospheric ozone thus equals the simulations with ozone treatments and results in an identical simulated biomass. We tested accounting for effects of ozone on stomatal conductance besides net photosynthesis as suggested by Lombardozzi et al. (2013). However, this additional direct damage to stomatal conductance yielded a minimal decrease in

simulated biomass accumulation in needle-leaved trees, but did not qualitatively change the results (results not shown). These results indicate that damage functions, with a large intercept and a very shallow (or non-existing) slope cannot simulate the impact of spatially varying $O_3$ concentrations or altered atmospheric $O_3$ concentrations.



The simulations L12$_{PS}$ and L12$_{VC}$ (net photosynthesis and $V_{cmax}$ damage according to Lombardozzi et al. (2012a) respectively) strongly overestimate biomass damage compared to Büker et al. (2015). Both damage functions assume an extensive damage to carbon fixation at low ozone accumulation values ($CUOY$) of about 5 $mmol\,O_3$. This results in a very steep decline in relative biomass at low values of $POD3$. Notably, despite a linear damage relationship, the very steep initial decline

in biomass of broadleaved trees at low values of $POD3$ is not continued at higher exposure, resulting in a non-linear biomass dose-response relationships. Higher accumulation of ozone doses does not result in higher damage rates beyond a threshold of about 5 $mmol\,O_3\,m^{-2}$ leaf area, and relative biomass declines remain 50 to 70 %. Whereas non-linear dose-response relationships are observed in experiments e.g. for leaf injury (Marzuoli et al., 2009), such a non-linear relationship is not produced in the biomass dose-response relationship by Büker et al. (2015).

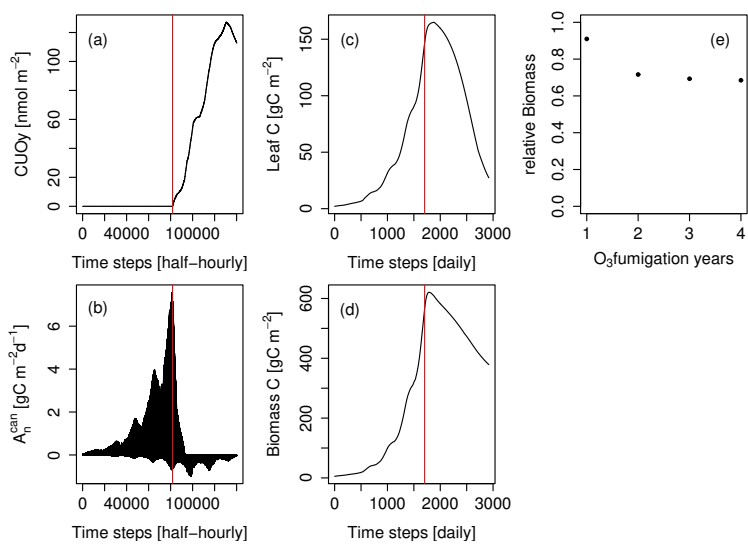

**Figure 2.** Cumulative ozone uptake above a threshold of 0.8 $nmol\,m^{-2}\,(leaf\,area)\,s^{-1}$ ($CUOY$), canopy integrated net photosynthesis ($A_n^{can}$), leaf carbon content ($Leaf\,C$), total carbon in biomass ($Biomass\,C$) and relative Biomass ($RB$) of *Pinus halepensis* at the Ebro Delta fumigated with the NF+ ozone treatment. Simulations are conducted with the L12$_{PS}$ model version. Panels a-d display the entire simulation period. The red line indicates the onset of $O_3$ fumigation (NF+) in the 5th of 8 simulations years. The relative biomass compared to a control simulation with zero $O_3$ concentration (panel e) is displayed for the $O_3$ fumigation years.

We investigated the cause for this at the example of the *Pinus halepensis* stand in the Ebro Delta with a high ozone treatment as shown in Fig. 2. The $CUOY$ quickly increases after the onset of fumigation (Fig. 2a) and is paralleled by a rapid decline in canopy integrated net photosynthesis ($A_n^{can}$, see Fig. 2b). Once all canopy layers accumulated more than 5 $mmol\,O_3$ $m^{-2}$, the canopy photosynthesis is fully reduced, and $A_n^{can}$ becomes negative as a consequence of ongoing leaf maintenance respiration. Thereafter, leaf and total biomass steadily decline (Fig. 2c,d), and the plants are kept alive only by the consumption

of stored non-structural carbon reserves. Despite the 100 % reduction in gross photosynthesis, the biomass compared to a control simulation (relative biomass, $RB$) reaches only values of approximately 0.7 (Fig. 2e), because of the remaining woody and root tissues (see Eq. 6 for the calculation of $RB$).





## 3.2 Tuned damage relationships

We next tested whether a linear damage function is in principle able to reproduce the observed biomass dose-response relationships. Simulations conducted with our tuned damage relationships produce biomass dose-response relationships which fit the target area defined by the $B_{SI}$ and $B_{ST}$ dose-response relationships by Büker et al. (2015) (see Fig. 3 and Tab. A.5, A.6).

For the calibrated relationships used in these simulations, we chose a flux threshold value of 1 $nmol\,m^{-2}(leaf\,area)\,s^{-1}$, as suggested by LRTAP-Convention (2017). We forced the intercept ($a$) of these relationships through 1, to simulate zero ozone damage at ozone fluxes lower than 1 $nmol\,m^{-2}(leaf\,area)\,s^{-1}$. The resulting slope of the $tun_{PS}$ function for broadleaved PFTs is approximately 30 times higher compared to the slope suggested by Wittig et al. (2007) and a fourth of the slope by Lombardozzi et al. (2012a). For the needle-leaved PFT, the tuned slope ($tun_{PS}$) is approximately 10 times higher (lower) than

the slopes by Wittig et al. (2007) and Lombardozzi et al. (2012a), respectively. Notably, we did not observe any difference in the model performance irrespective of whether net photosynthesis or photosynthetic capacity ($V_{cmax}$) was reduced.

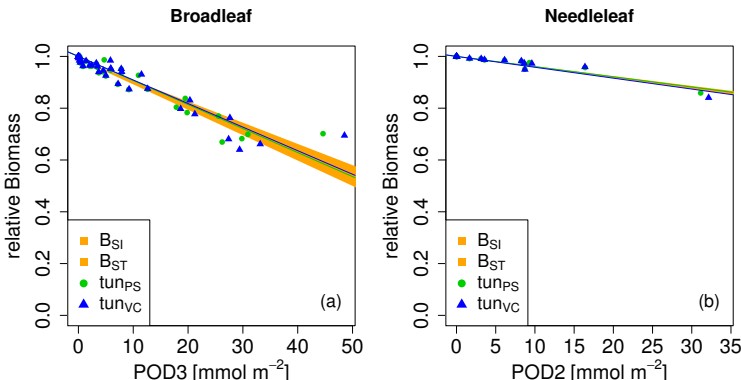

**Figure 3.** Biomass dose-response relationships for simulations based on tuned damage functions (see Tab. 1 for abbreviations), separate for a) broadleaved species, and b) needle-leaved species. The dose-response relationships by Büker et al. (2015), $B_{SI}$ and $B_{ST}$, define the target area (orange). See Tab. A.5 and A.6 for slopes, intercepts, $R^2$ and p-values of the displayed regression lines. Damage calculation in the simulations base on $CUO1$ (see Tab. 1) and not on $POD2$ or $POD3$ (see Sec. 2.4 for more details).

## 3.3 Ozone damage to mature trees

The simulation of young trees (simulated as in the previous section) compared to adult trees with the same model version reveals a distinct difference between the simulated versus observed dose-response relationship when expressed as reduction of

15 biomass. Ozone damage causes a much shallower simulated biomass dose-response relationship for adult trees ($tun_{VC}^{mature}$ in Fig. 4a,b) compared to young trees ($tun_{VC}^{young}$ in Fig. 4a,b), both for broadleaved and needle-leaved species. It is worth noting that this is primarily the consequence of the higher initial biomass of the adult trees before ozone fumigation starts ($tun_{VC}^{mature}$). Comparing the dose-response relationship of young and mature trees based on the annual net biomass production (NPP) shows nearly identical slopes for needle-leaved species (Fig. 4d and Tab. 3), whereas the slopes for broadleaved tree species (Fig. 4c





and Tab. 2) suggests only a slightly lower reduction in NPP in mature compared to young trees, likely related to the larger amount of non-structural reserves that increases the resilience of mature versus young trees.

**Table 2.** Slopes and intercepts of biomass (RB) and NPP (RN) dose-response relationships (DRR) for broadleaved species simulated by the $\text{tun}_{VC}$ model version (see Tab. 1). The fumigation of young trees ($\text{tun}_{VC}^{young}$) with $O_3$ is compared to the fumigation of mature trees ($\text{tun}_{VC}^{mature}$).

| DRR | ID | Intercept (a) | Slope (b) | $R^2$ | p-value |
|-----|-----|-----|-----|-----|-----|
| RB | $\text{tun}_{VC}^{young}$ | 1 | 0.0091 | 0.93 | 5e-25 |
| RB | $\text{tun}_{VC}^{mature}$ | 1 | 0.00142 | 0.91 | 9.8e-23 |
| RN | $\text{tun}_{VC}^{young}$ | 1 | 0.0167 | 0.96 | 6.2e-30 |
| RN | $\text{tun}_{VC}^{mature}$ | 1 | 0.0144 | 0.93 | 1.4e-24 |

**Table 3.** Slopes and intercepts of biomass (RB) and NPP (RN) dose-response relationships (DRR) for needle-leaved species simulated by the $\text{tun}_{VC}$ model version (see Tab. 1). The fumigation of young trees ($\text{tun}_{VC}^{young}$) with $O_3$ is compared to the fumigation of mature trees ($\text{tun}_{VC}^{mature}$).

| DRR | ID | Intercept (a) | Slope (b) | $R^2$ | p-value |
|-----|-----|-----|-----|-----|-----|
| RB | $\text{tun}_{VC}^{young}$ | 1 | 0.0042 | 0.93 | 2.2e-09 |
| RB | $\text{tun}_{VC}^{mature}$ | 1 | 0.000785 | 0.79 | 4.2e-06 |
| RN | $\text{tun}_{VC}^{young}$ | 1 | 0.00858 | 0.97 | 2.3e-12 |
| RN | $\text{tun}_{VC}^{mature}$ | 1 | 0.00808 | 0.99 | 3 .7e-16 |

## 4 Discussion

Damage functions that relate accumulated ozone uptake to fundamental plant processes such as photosynthesis are a key component for models that aim to estimate the potential impacts of ozone pollution on forest productivity, growth and carbon sequestration. We tested four published damage functions for net photosynthesis and $V_{cmax}$ within the framework of the O-CN model to assess their ability to reproduce the empirical whole tree biomass dose-response relationships derived by Büker et al. (2015). The biomass dose-response relationships calculated from the O-CN simulations show that the parameterisation of the damage functions included in the model have a large impact on the simulated whole tree biomass: the published damage functions either substantially over- or substantially under-estimated whole tree biomass reduction compared to the data presented by Büker et al. (2015).

The simulation results from the O-CN version applying a damage function based on a single, ozone-sensitive species (Lombardozzi et al., 2012a) to a range of European tree species leads to a strong overestimation of the simulated biomass damage




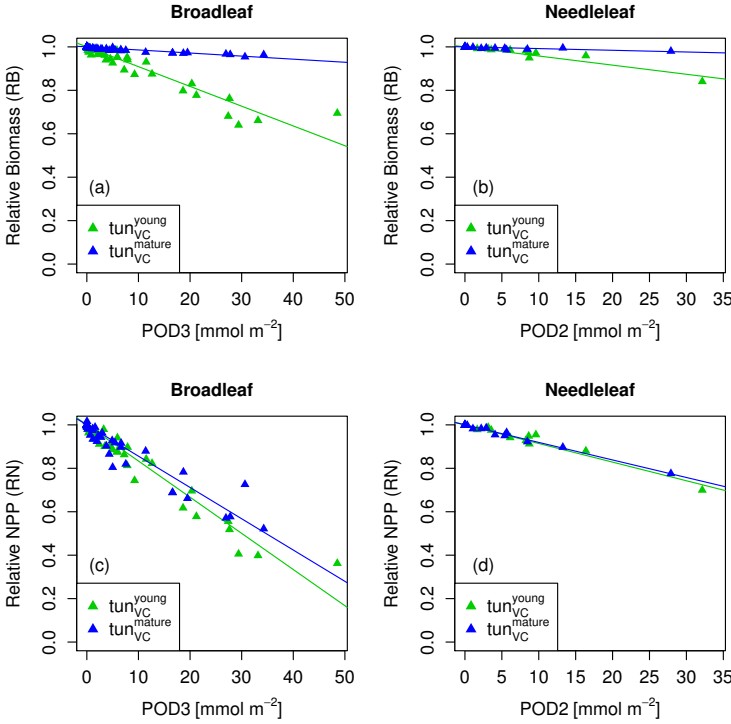

**Figure 4.** Biomass (RB) and NPP (RN) dose-response relationships of simulations with young (tun$_{VC}^{young}$) and mature trees (tun$_{VC}^{mature}$) separate for a,c) Broadleaf species, and b,d) Needleleaf species.

compared to the observations used in this study. The problem of using such damage parameterisations based on short-term experiments of ozone-sensitive species is further highlighted when applying them in simulations of multiple season fumigation experiments and/or high ozone concentrations. Under such conditions, fumigation with high $O_3$ concentrations can lead to lethal doses, which might not be observed in field experiments due to restricted experiment lengths. Previous studies have

suggested that in large areas of Europe, the Eastern US and South-East Asia average growing season values of $CUOY$ for recent years range between 10-100 $mmol\ O_3\ m^{-2}$ (Lombardozzi et al., 2015; Franz et al., 2017). The damage relationships L12$_{PS}$ and L12$_{VC}$ by Lombardozzi et al. (2012a) assume a 100% damage to net photosynthesis or $V_{cmax}$ at accumulation values of about 5 $mmol\ O_3\ m^{-2}$. This would imply that in these large geographic regions, photosynthesis would have been completely impaired by ozone, which is clearly not the case. This result highlights the need for a representative set of species

for the development of damage functions for large-scale biosphere models. Overall, our results suggests that the estimates of global GPP reduction as a result of ozone pollution by Lombardozzi et al. (2012a) are strongly overestimated.

Meta-analyses (Wittig et al., 2007; Lombardozzi et al., 2013) are designed to minimise the effect of species-specific ozone sensitivities and provide estimates of the average species response. However, we found that the relationships derived by these meta-analyses substantially underestimate biomass damage. Technically, the reasons for this are a weak or non-existent in-

crease of the ozone damage with increased ozone uptake (shallow or non-existent slopes) and/or high ozone damage at zero





accumulated ozone uptake (intercept lower than one). Apparently, the diversity of species responses and experimental settings that are assembled in the meta-analyses by Wittig et al. (2007) and Lombardozzi et al. (2013), together with uncertainties in precisely estimating accumulated ozone uptake in these databases preclude the identification of damage functions that are consistent with the damage estimates by Büker et al. (2015). The high intercepts in the meta-analyses by Wittig et al. (2007)

and Lombardozzi et al. (2013), which assume a considerable damage fraction even when no ozone is taken up at all, seem to be ecologically illogical and suggest that an alternative approach is necessary to simulate ozone damage. As a consequence of these points, the Europe-wide GPP reduction estimates by Franz et al. (2017), which has been based on the damage function by Wittig et al. (2007), may substantially underestimate actual GPP reduction. Similarly, global estimates as well as spatial variability of ozone damage to GPP by Lombardozzi et al. (2015), based on Lombardozzi et al. (2013), are virtually independent

of actual ozone concentrations or uptake for all tree plant functional types and should be interpreted with caution.

A crucial aspect in forming dose-response relationships is the calculation of the accumulated ozone uptake (e.g. $PODy$ or $CUOY$). The calculation of accumulated ozone uptake is realised in different ways in the meta-analyses and the study by Büker et al. (2015) as well as in our approach here. Experiments synthesised in the meta-analyses generally do not have access to stomatal conductance values at high resolution measured throughout the experiment, which impedes precise determination of

$O_3$ uptake. The uncertainty in the necessary approximations of accumulated ozone uptake can be assumed to be considerable, and it is thus highly recommendable to measure and report required observations in future ozone fumigation experiments. Büker et al. (2015) use the DO$_3$SE model to simulate ozone uptake and accumulation similar as done in our model here. These modelled values for ozone uptake and accumulation can assumed to be more reliable since both models simulate processes that determine ozone uptake continuously for the entire experiment length at high temporal resolution. They account for diurnal

changes in stomatal conductance as well as climate factors restricting stomatal conductance and hence ozone uptake. However, both models vary in their complexity of the simulated plants, carbon assimilation, and growth processes, which will also impact the estimate of ozone accumulation.

The meta-analyses do not account for non-stomatal ozone deposition (e.g. to the leaf cuticle or soil), which imposes a bias towards overestimating ozone uptake and accumulation contrary to the DO$_3$SE model used by Büker et al. (2015), which

accounts for this. The O-CN model in principle can simulate non-stomatal ozone deposition from the free atmosphere to ground level (see Franz et al. (2017)). The leaf boundary layer is implicitly included into the calculation of the aerodynamic resistance of O-CN and included in Franz et al. (2017). However, for the simulation of the chamber experiments we used the observed chamber $O_3$ concentrations, rather than estimating the canopy-level $O_3$ concentration based on the free atmosphere (approximately 45 $m$ above the surface) and atmospheric turbulence. This required not accounting for aerodynamic resistance

and therefore the leaf-boundary layer resistance as well as it prevented the calculation of the non-stomatal deposition, which may lead to a slight overestimation of ozone uptake and accumulation in our simulations.

The calibration of damage functions to net photosynthesis and V$_{cmax}$ shows that in principle, the linear structure of Eq. 4 is sufficient to simulate biomass dose-response relationships comparable to Büker et al. (2015) in O-CN. An advantage of the damage functions derived here compared to previously published damage functions (Wittig et al., 2007; Lombardozzi

et al., 2012a, 2013) is the intercept of one, implying that simulated ozone damage is zero at zero accumulated $O_3$ and steadily





increases with increased ozone accumulation. The flux threshold used in the simulations is $1\ nmol\ m^{-2}(leaf\ area)\ s^{-1}$ as suggested by the LRTAP-Convention (2017). Since the tuned damage functions are structurally identical to previously published damage functions based on accumulated ozone uptake they can be directly compared to them. Slopes of the tuned damage functions lie in between the values proposed by Wittig et al. (2007) and Lombardozzi et al. (2012a) and thus take values in an

expected range. We did not find any significant difference in simulated biomass responses between the use of net photosynthesis or leaf-specific photosynthetic capacity ($V_{cmax}$) as a target for the ozone damage function, although we do note that the slopes were slightly lower for the net photosynthesis based functions. The simulation of ozone effects on leaf-specific photosynthetic capacity ($V_{cmax}$) seems preferable over the adjustment of net photosynthesis, because $V_{cmax}$ and $J_{max}$ are parameters in the calculation of net photosynthesis, and thus are likely easier transferable between models. Models with different approaches to

simulate net photosynthesis might obtain better comparable results by using damage relationships that target $V_{cmax}$ instead of net photosynthesis.

All damage functions included into the O-CN model base damage calculations on the damage index $CUOY$ (canopy value) rather than $PODy$, as used by some other models, e.g. the DO$_3$SE model (Emberson et al., 2000). We tested the effect of basing the damage calculation on $POD1$ rather than $CUO1$, and found that these produced comparable biomass dose-response

relationships as the damage relationships based on $CUO1$ presented in Fig. 3 (results not shown). The slopes of damage functions based on $POD1$ are approximately two thirds and half compared to the slopes based on $CUO1$ for broadleaved and needle-leaved species respectively. The difference in the slope values associated with $POD1$ and $CUO1$ results from the different calculation and application of them. The $POD1$ is calculated in the top canopy layer and the respective damage fraction is applied for all canopy layers, the $CUOY$ though is calculated separately in each canopy layer as well as the

respective damage fraction. Higher frequency data on the ozone damage incurred by plants are required to disentangle whether an ozone damage parameterisation based on instantaneous or accumulated ozone uptake results in more accurate simulation of the seasonal effects and more analysis of the differential effect of ozone damage within deep canopies are required to evaluate whether the scaling of top-of-the-canopy damage to whole canopy damage is appropriate.

Further aspects that determine ozone sensitivity and damage to carbon gain of plants like leaf morphology (Calatayud

et al., 2011; Bussotti, 2008), different sensitivity of sunlit and shaded leafs (Tjoelker et al., 1995; Wieser et al., 2002), early senescence (Gielen et al., 2007; Ainsworth et al., 2012) and costs for detoxification of ozone and/or repair of ozone damage that likely increases the plant's respiration costs (Dizengremel, 2001; Wieser and Matyssek, 2007) are not considered by either approach. Marzuoli et al. (2016) observed an ozone induced reduction of biomass but no significant reduction in physiological parameters like $V_{cmax}$. They suggest that the reduced growth is caused by higher energy investments and reducing power for

the detoxification of ozone whereas the photosynthetic apparatus remained undamaged (Marzuoli et al., 2016).

Some studies have found that ozone-affected stomata respond much slower to environmental stimuli than unaffected cells (Paoletti and Grulke, 2005), which can delay closure and trigger, stomatal sluggishness, an uncoupling of stomatal conductance and photosynthesis (Reich, 1987; Tjoelker et al., 1995; Lombardozzi et al., 2012b) and thus impact transpiration rates (Mills et al., 2009; Paoletti and Grulke, 2010; Lombardozzi et al., 2012b) and the plant's water use efficiency (Wittig et al., 2007;

Mills et al., 2009; Lombardozzi et al., 2012b). The O-CN model is able to directly impair stomatal conductance, by uncoupling





damage to net photosynthesis from the subsequent damage to stomatal conductance. In this version of the O-CN model both net photosynthesis and stomatal conductance can directly be damaged by individual damage functions. The simulation of this kind of direct damage to stomatal conductance additional to the damage of net photosynthesis, both according to the damage functions by (Lombardozzi et al., 2013), have a negligible impact on biomass production compared to not accounting for direct

damage to the stomata (results not shown). However, our above mentioned concerns regarding the structure of the damage relationships by Lombardozzi et al. (2013) should be taken into account when considering this result.

A key challenge for the use of fumigation experiments to parameterise ozone-damage in models is that trees (as opposed to grasses fumigated from seeds) typically possess a certain amount of biomass at the beginning of the fumigation experiment. Even at lethal ozone doses, the relative biomass thus can not decline to zero, and tree death may occur at values of a relative

biomass greater than zero. The relative biomass is positive even if carbon fixation is fully reduced and the plants survive due to the use of stored carbon. The higher the initial biomass and the slower the annual biomass growth rate of the tree is, the harder it is to obtain low values of $RB$. When comparing $RB$ values obtained from trees with substantial different initial biomass and tree species with different growth rates proportionate damage rates thus can not directly be inferred. This indicates that the explanatory value of the relative biomass between a control and a treatment to estimate long-term plant damage at a given

$O_3$ concentration is limited. This is particularly the case when evaluating the damage of more mature forests. The simulated biomass dose-response relationships of adult trees are much more shallow than dose-response relationships of young trees (see Fig. 4), because of the high initial biomass prior to fumigation. This suggests that the use of biomass damage functions derived from experiments with young trees to parameterise the biomass loss of adult trees, as done in Sitch et al. (2007), will likely lead to an overestimation of plant damage and loss of carbon storage. Dose-response relationships based on biomass

increments or growth rates might be better transferable between saplings and mature trees and hence better suitable to be used for parameterising global terrestrial biosphere models.

Our approach to overcome this challenge was to alter the vegetation model to simulate the ozone damage of small trees, where we could directly compare simulated biomass reductions to observations. Since we used damage relationships that are based on the calculation of leaf-level photosynthesis, we are able to apply the calibrated model also for mature stands. Our

simulations have demonstrated that despite the different sizes of young and mature trees, and associated changes in the wood growth rate and the available amount of non-structural carbon reserves to repair incurred damage, the simulated effect of ozone on the net annual biomass production (NPP) was very similar, when using a damage function associated with leaf-level photosynthesis. Overall our findings support the idea that the photosynthesis-based damage relationships developed here and evaluated against fumigation experiments of young trees, might be useful to estimate effect on forest production of older trees.

Monitoring approaches that are either capable of measuring the actual increment of biomass, or quantify at the leaf and canopy level the change in net photosynthesis over the growing season, would allow to develop damage estimates that could be more readily translated into modelling frameworks.

Terrestrial biosphere models in general assume that plant growth is primarily determined by carbon uptake. However, an alternative concept proposes that plant growth is more limited by direct environmental controls (temperature, water and nutrient

availability) than by carbon uptake and photosynthesis (Fatichi et al., 2014). The O-CN model provides a first step into this





direction because it separates the step of carbon acquisition from biomass production, both in terms of a non-structural carbon buffer, as well as a stoichiometric nutrient limitation on growth independent of the current photosynthetic rate. This would in principle allow to account for ozone effects on the carbon sink dynamics within plants. However, it is not clear that data readily exist to parameterise such effects. Given the availability of suitable data to parameterise a large scale model, ozone damage

might be better simulated by targeting biomass growth rates or processes that limit these e.g. stomatal conductance, which impacts the plants water balance compared to our approach here, which targets net photosynthesis.

All in all a multitude of aspects that impact ozone damage to plants is not yet incorporated into global terrestrial biosphere models. The ongoing discussion which processes are major drivers for observed damages, how they interact and impact different species and plant types plus the lack of suitable data needed to parameterise a global model are reasons why the simulation

of ozone damage up to now focuses only on a few aspects where suitable data are available as presented in our study.

## 5   Conclusion

The inclusion of previously published dose-response relationships into the terrestrial biosphere model O-CN led to a strong over- or underestimation of simulated biomass damage compared to the biomass dose-response relationship by Büker et al. (2015). Dose-response relationships which are used as damage functions in terrestrial biosphere models are a key aspect in the

simulation of ozone damage and have a great impact on the estimated damage. The calibration of damage functions performed in this study provide the advantage to calculate ozone damage close to where the actual physiological damage might occur (photosynthetic apparatus) and simultaneously reproduce observed biomass damage relationships for a range of European forest species used by Büker et al. (2015). The inclusion of these damage functions into models that estimate regional or global ozone damage might improve damage estimates compared to previously published damage functions and might lead to better

estimates of terrestrial carbon sequestration. The comparison of simulated biomass dose-response relationships of young and mature trees shows strongly different slopes. This suggests that observed biomass damage relationships from young trees might not be suitable to estimate biomass damage of mature trees. The comparison of simulated NPP dose-response relationships of young and mature trees show similar relationships and suggests that they might more readily be transferred between trees differing in age.

*Acknowledgements.* We would like to thank Per Erik Karlsson of the IVL Swedish Environmental Research Institute, Göteborg, Sweden, Sabine Braun of the Institute for Applied Plant Biology, Witterswil, Switzerland, and Gerhard Wieser of the Federal Research and Training Centre for Forests, Natural Hazards and Landscape (BFW), Innsbruck, Austria for providing collected data from their ozone fumigation experiments. This research leading to this publication was supported by the EU Framework programme through grant no. 282910 (ECLAIRE), and the Max Planck Society for the Advancement of Science e.V. through the ENIGMA project. This project has received funding from

the European Research Council (ERC) under the European Union's Horizon 2020 research and innovation programme (grant agreement no. 647204; QUINCY).



**Table A.1.** Original and adapted values of the nitrogen specific photosynthetic capacity of a leaf (npl) for three out of four different O-CN versions (ID) including published damage functions. The intercept of the fourth O-CN version (L12$_{VC}$) is very close to one and simulations produce comparable LAI values without an adaption of npl.

| ID | PFT | npl original | npl adapted |
|---|---|---|---|
| W07$_{PS}$ | Broadleaf | 1.50 | 1.60 |
| W07$_{PS}$ | Needleleaf | 0.75 | 0.80 |
| L12$_{PS}$ | Broadleaf | 1.50 | 1.45 |
| L12$_{PS}$ | Needleleaf | 0.75 | 0.70 |
| L13$_{PS}$ | Broadleaf | 1.50 | 1.75 |
| L13$_{PS}$ | Needleleaf | 0.75 | 0.90 |

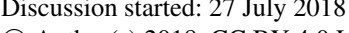



**Table A.2.** List of fumigation experiments used by Büker et al. (2015) and simulated here.

| Site | Longitude [°E] | Latitude [°N] | Species | $O_3$ treatment start year | Fumigation [yrs] |
|---|---|---|---|---|---|
| Östad (S) | 12.4 | 57.9 | *Betula pendula* | 1997 | 2 |
| Birmensdorf (CH) | 8.45 | 47.36 | *Betula pendula* | 1989 | 1 |
| Birmensdorf (CH) | 8.45 | 47.36 | *Betula pendula* | 1990 | 1 |
| Birmensdorf (CH) | 8.45 | 47.36 | *Betula pendula* | 1992 | 1 |
| Birmensdorf (CH) | 8.45 | 47.36 | *Betula pendula* | 1993 | 1 |
| Kuopio (FIN) | 27.58 | 62.21 | *Betula pendula* | 1994 | 2 |
| Kuopio (FIN) | 27.58 | 62.21 | *Betula pendula* | 1996 | 3 |
| Kuopio (FIN) | 27.58 | 62.21 | *Betula pendula* | 1994 | 5 |
| Schönenbuch (CH) | 7.5 | 47.54 | *Fagus sylvatica* | 1991 | 2 |
| Zugerberg (CH) | 8.54 | 47.15 | *Fagus sylvatica* | 1987 | 2 |
| Zugerberg (CH) | 8.54 | 47.15 | *Fagus sylvatica* | 1989 | 3 |
| Zugerberg (CH) | 8.54 | 47.15 | *Fagus sylvatica* | 1991 | 2 |
| Curno (I) | 9.03 | 46.17 | *Populus spec.* | 2005 | 1 |
| Grignon (F) | 1.95 | 48.83 | *Populus spec.* | 2008 | 1 |
| Ebro Delta (SP) | 0.5 | 40.75 | *Quercus ilex* | 1998 | 3 |
| Col-du-Donon (F) | 7.08 | 48.48 | *Quercus robur or petraea* | 1999 | 2 |
| Headley (U.K.) | -0.75 | 52.13 | *Quercus robur or petraea* | 1997 | 2 |
| Ebro Delta (SP) | 0.5 | 40.75 | *Pinus halepensis* | 1993 | 4 |
| Col-du-Donon (F) | 7.08 | 48.48 | *Pinus halepensis* | 1997 | 2 |
| Schönenbuch (CH) | 7.5 | 47.54 | *Picea abies* | 1991 | 2 |
| Zugerberg (CH) | 8.54 | 47.15 | *Picea abies* | 1991 | 2 |
| Östad (S) | 12.4 | 57.9 | *Picea abies* | 1992 | 5 |
| Headley (U.K.) | -0.75 | 52.13 | *Pinus sylvestris* | 1995 | 2 |



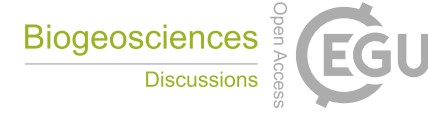

**Table A.3.** Slopes and intercepts of biomass dose-response relationships for broadleaved species simulated by O-CN versions based on published damage functions to net photosynthesis or $V_{cmax}$ (see Tab. 1). $B_{SI}$ and $B_{ST}$ represent the simple and standard model of Büker et al. (2015).

| ID | Intercept (a) | Slope (b) | $R^2$ | p-value |
|----|----|----|----|----|
| $B_{SI}$ | 0.99 | 0.0082 | 0.34 | <0.001 |
| $B_{ST}$ | 0.99 | 0.0098 | 0.38 | <0.001 |
| $W07_{PS}$ | 1 | 0.00045 | 0.93 | 1e-24 |
| $L12_{PS}$ | 1 | 0.0142 | 0.77 | 2e-14 |
| $L15_{PS}$ | 1 | 0.0000 | NaN | NaN |
| $L12_{VC}$ | 1 | 0.0120 | 0.80 | 1.9e-15 |

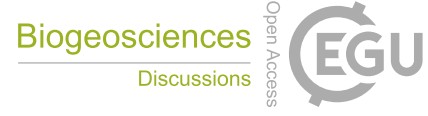

**Table A.4.** Slopes and intercepts of biomass dose-response relationships for needle-leaved species simulated by O-CN versions based on published damage functions to net photosynthesis or $V_{cmax}$ (see Tab. 1). $B_{SI}$ and $B_{ST}$ represent the simple and standard model by Büker et al. (2015).

| ID | Intercept (a) | Slope (b) | $R^2$ | p-value |
|----|---------------|-----------|-------|---------|
| $B_{SI}$ | 1 | 0.0038 | 0.46 | <0.001 |
| $B_{ST}$ | 1 | 0.0042 | 0.52 | <0.001 |
| $W07_{PS}$ | 1 | 0.00058 | 0.93 | 1.5e-09 |
| $L12_{PS}$ | 1 | 0.0119 | 0.83 | 9.4e-07 |
| $L15_{PS}$ | 1 | 0.0000 | NaN | NaN |
| $L12_{VC}$ | 1 | 0.0096 | 0.85 | 3.5e-07 |

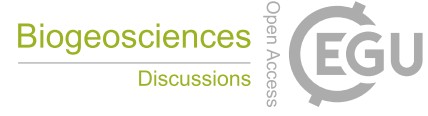

**Table A.5.** Slopes and intercepts of biomass dose-response relationships for broadleaved species simulated by O-CN versions based on tuned damage functions to net photosynthesis or $V_{cmax}$ (see Tab. 1). $B_{SI}$ and $B_{ST}$ represent the simple and standard model by Büker et al. (2015).

| ID | Intercept (a) | Slope (b) | $R^2$ | p-value |
|----|---------------|-----------|-------|---------|
| $B_{SI}$ | 0.99 | 0.0082 | 0.34 | <0.001 |
| $B_{ST}$ | 0.99 | 0.0098 | 0.38 | <0.001 |
| $tun_{PS}$ | 1 | 0.0093 | 0.94 | 1.4e-26 |
| $tun_{VC}$ | 1 | 0.0091 | 0.93 | 5e-25 |



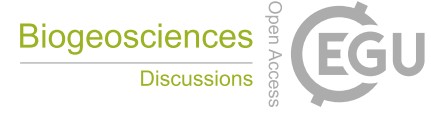

**Table A.6.** Slopes and intercepts of biomass dose-response relationships for needle-leaved species simulated by O-CN versions based on tuned damage functions to net photosynthesis or $V_{cmax}$ (see Tab. 1). $B_{SI}$ and $B_{ST}$ represent the simple and standard model by Büker et al. (2015).

| ID | Intercept (a) | Slope (b) | $R^2$ | p-value |
|---|---|---|---|---|
| $B_{SI}$ | 1 | 0.0038 | 0.46 | <0.001 |
| $B_{ST}$ | 1 | 0.0042 | 0.52 | <0.001 |
| $\text{tun}_{PS}$ | 1 | 0.0039 | 0.94 | 4.8e-10 |
| $\text{tun}_{VC}$ | 1 | 0.0042 | 0.93 | 2.2e-09 |



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
