# Peer review of "Evaluation of simulated biomass damage in forest ecosystems induced by ozone against observation-based estimates"

_Biogeosciences, 2018_

## Referee Comment (RC1) · Anonymous Referee #1 · 26 Aug 2018

Using the O-CN vegetation model as a testbed, the authors specifically tested four ozone damage functions in terms of simulated biomass reduction against measured ozone dose-biomass response data across Europe and finally derived tuned damage functions that can much better simulate the biomass responses.

This work is a very well-designed modelling study. Modelling protocols and results are clearly presented. Overall this manuscript is very well written and easy to follow most of the time, which should be accepted for formal publication after a minor revision.

However, I do have a few questions that may need the authors to further clarify:

1)The authors assume that the modelled accumulation of ozone fluxes at the top

canopy layer equals POD during the model-observation comparison process. Please justify this assumption. I think this is important for the evaluation of model against observation, considering the ozone damage is explicitly calculated through the canopy and integrated to derive the whole tree damage. The modelled POD value largely influences the slope of the resultant dose-response curve and its distance with observed dose-response curve. I am wondering how would the authors account for this treatment in influencing the evaluation of different algorithms against observed data.

2) I am curious why did not the author try to use different damage functions at different depth of the canopy?

3) Another important, but still largely missing, aspect in simulating ozone impacts on vegetation is the huge diversity of species-sensitivity in an ecosystem. Dealing with vegetation to the PFT level is not enough, though totally make sense in terms of large scale modelling and data scarcity. This work could be improved by further talking about diversity of species response to ozone. To this end, I found the following work could be a good reference: Wang, B. et al. Forests and ozone: productivity, carbon storage, and feedbacks. Sci. Rep. 6, 22133; doi: 10.1038/srep22133 (2016)

This study, though without sophisticated ozone damage simulation, had an explicit simulation of species sensitivity to ozone using an individual-based model and found dampened responses to ozone over long-term simulations. Minor comments

L27 on page 4: please justify the statement of highest N concentration at the top of the canopy and its exponential decline with increasing canopy depth.

L18 on page 5: in equation 2, how is the stomatal conductance of O3 calculated?

L28 on page 8: identical -> identically.

L5-6 on Page 18: this sentence should be restructured to make it easier to follow.

L7 on page 18: ''all in all' should be followed a comma.

---

## Author Comment (AC1) · 5 Sep 2018

**1   Answers to Anonymous Referee # 1**

Q: The authors assume that the modelled accumulation of ozone fluxes at the top canopy layer equals POD during the model-observation comparison process. Please justify this assumption. I think this is important for the evaluation of model against observation, considering the ozone damage is explicitly calculated through the canopy and integrated to derive the whole tree damage. The modelled POD value largely influences the slope of the resultant dose-response curve and its distance with observed

dose-response curve. I am wondering how would the authors account for this treatment in influencing the evaluation of different algorithms against observed data.

A: We designed our study such that our way to calculate $POD$ is consistent with those from Büker et al. (2015), from which we took the dose-response-relationships. They calculate the $PODy$ used for their analysis in accordance with the LRTAP-Convention (2010), which states 'the index PODY is used to quantify the flux of ozone through the stomata of the uppermost leaf level that is directly exposed to solar radiation and thus no calculation of light exclusion, caused by the filtering of light through the leaves of the canopy, is required'. We calculate the $PODy$ based on the LRTAP-Convention (2010), to be able to compare our simulation results to those of Büker et al. (2015).

We will add a citation of the LRTAP convention to the explanation on the calculation of POD in the text: 'For comparison to observations, the Phytotoxic Ozone Dose ($POD$, $mmol m^{-2}$) can be diagnosed by the accumulation of $f_{st,l}$ for the top canopy layer (l = 1), in accordance with LRTAP-Convention (2010) and Büker et al. (2015).'

Of course, there will be uncertainty in the calculation of the $POD$ by both Büker et al. (2015) and our study compared to the real-world $POD$, given both are based on different, but evaluated models (Emberson et al., 2000; Franz et al., 2017), but in the absence of direct measurements of $POD$ it is impossible to judge whether or not this would introduce any systematic bias into the comparison.

Q: I am curious why did not the author try to use different damage functions at different depth of the canopy?

A: Each of the damage functions is applied to all canopy layers in separate simulations for each damage function. The ozone damage differs within the canopy, as increasing canopy depth leads to lower leaf-specific photosynthesis, conductance, and therefore ozone uptake and damage.

Our aim was to investigate the suitability of different damage functions to reproduce observed biomass damage relationships. Following this we always only applied one damage function in one simulation. The application of different damage functions in one simulation, e.g. different damage functions for different canopy layers, can not contribute to answer our research question.

Evidence exists that sunlit and shades leaves exhibit a different sensitivity to $O_3$ (Tjoelker et al., 1995; Wieser et al. 2002). Following this the application of different damage functions for different canopy layers might yield improved damage estimates. However damage relationships for different canopy depth are to our knowledge not available as well as independent data to evaluate them.

Q: 3) Another important, but still largely missing, aspect in simulating ozone impacts on vegetation is the huge diversity of species-sensitivity in an ecosystem. Dealing with vegetation to the PFT level is not enough, though totally make sense in terms of large scale modelling and data scarcity. This work could be improved by further talking about diversity of species response to ozone. To this end, I found the following work could be a good reference: Wang, B. et al. Forests and ozone: productivity, carbon storage, and feedbacks. Sci. Rep. 6, 22133; doi: 10.1038/srep22133 (2016)

This study, though without sophisticated ozone damage simulation, had an explicit simulation of species sensitivity to ozone using an individual-based model and found dampened responses to ozone over long-term simulations.

A: We will include this study in our discussion.

**1.1 Minor comments**

Q: L27 on page 4: please justify the statement of highest N concentration at the top of the canopy and its exponential decline with increasing canopy depth.
A: We base this statement on the publications by Friend (2001) and Niinemets et al. (2015) and will add these references in the manuscript.

Q: L18 on page 5: in equation 2, how is the stomatal conductance of O3 calculated?
A: Explanation added.

Q: L28 on page 8: identical →identically.
A: Done.

Q: L5-6 on Page 18: this sentence should be restructured to make it easier to follow.
A: Done.

Q: L7 on page 18: ''all in all' should be followed a comma.
A: Done.

**2 References**

Büker, P., Feng, Z., Uddling, J., Briolat, A., Alonso, R., Braun, S., Elvira, S., Gerosa, G., Karlsson, P., Le Thiec, D., Marzuoli, R., Mills, G., Oksanen, E., Wieser, G., , Wilkinson, M., and Emberson, L.: New flux based dose-response relationships for ozone for European for- est tree species, Environmental Pollution, 206, 163–174, https://doi.org/

10.1016/j.envpol.2015.06.033, URL http://tinyurl.sfx.mpg.de/ug3q, 2015.

Emberson, L., Simpson, D., Tuovinen, J., Ashmore, M., and Cambridge, H.: Towards a model of ozone deposition and stomatal uptake over Europe, EMEP MSC-W Note, 6, 1–57, 2000.

Franz, M., Simpson, D., Arneth, A., and Zaehle, S.: Development and evaluation of an ozone deposition scheme for coupling to a terrestrial biosphere model, Biogeosciences, 14, 45–71, https://doi.org/10.5194/bg-14-45-2017, URL http://www.biogeosciences.net/14/45/2017/, 2017.

Friend, A.: Modelling canopy CO2 fluxes: are 'big-leaf'simplifications justified?, Global Ecology and Biogeography, 10, 603–619, 2001. LRTAP-Convention: Manual on Methodologies and Criteria for Modelling and Mapping Critical Loads and Levels; and Air Pollution Effects, Risks and Trends, http://www.rivm.nl/en/themasites/icpmm/index.html, URL http://www.rivm.nl/en/themasites/icpmm/index.html, 2010.

Niinemets, Ü, Keenan, T. F., and Hallik, L.: A worldwide analysis of within-canopy variations in leaf structural, chemical and physiological traits across plant functional types, New Phytologist, 205, 973–993, 2015.

Tjoelker, M., Volin, J., Oleksyn, J., and Reich, P.: Interaction of ozone pollution and light effects on photosynthesis in a forest canopy experiment, Plant, Cell  Environment, 18, 895–905, 1995.

Wieser, G., Hecke, K., Tausz, M., Haberle, K., Grams, T., and Matyssek, R.: The role of antioxidative defense in determining ozone sensitivity of Norway spruce (Picea abies (L.) Karst.)  across tree age: Implications for the sun-and shade-crown, PHYTON-HORN-, 42, 245–254, 2002.

---

## Referee Comment (RC2) · M. Schaub (Referee) · 6 Sep 2018

Review comments by Marcus Schaub, Maxime Cailleret and Marco Ferretti

The authors argue that so far applied damage functions result in impacts with large uncertainty in the magnitude of ozone effects predicted. They use the O-CN biosphere model to test four already existing damage functions in terms of their simulated whole-tree biomass responses against field data from 23 ozone filtration/fumigation experiments and found that biomass damage was overestimated (Lambardozzi et al. 2012) or underestimated (Wittig et al 2007; Lambardozzi et al. 2013). The authors tune/re-parameterize those damage functions towards a better fit with data from 15 fumigation

experiments with young trees. In a second step, the authors tune DRRs again so that relative biomass (or NPP) simulated on adult trees fit the measured values on young trees.

The ms. reads very well and is certainly within the scope of BG. As a matter of fact, we appreciate this exercise as it addresses a crucial issue in ozone risk assessment and provides an excellent review on the state of the art.

While the first part, i.e. recalibration of existing damage functions makes sense to improve DRRs for young trees and better predict biomass loss due to ozone. We are, however, concerned about the second step, i.e. the reparameterization/tuning of those functions (for young trees) to better predict relative biomass for mature trees. The authors aim at improving the quantitative understanding of ozone effects on forest growth and carbon sequestration on a regional or even global scale. Using data from seedlings gown under (semi-)controlled experiments ranging over a few years may (still) not lead to reliable model functions for adult trees growing in complex forest ecosystems. The cited work by Franz et al. 2017 (GPP reduction, based on damage functions from Wittig et al. 2007) is an example how model exercises using modeled data may result in inaccurate predictions – if not validated with measured data from adult trees (see also Cailleret et al. 2018). Page 15, line 6-10 demonstrates the risk of applying models, based on former functions and stresses the need of validating model exercises with measured data (e.g. from ICP Forests). We suggest to either omit the second part or to extend section 3.3. and the discussion and to outline not only the advances but also the still existing lack of knowledge for estimating ozone induced biomass effects on adult trees, forest ecosystems respectively.

Novak et al. (2008) found that species competition may alter DDRs. We did not understand if and how competition is considered in the O-CN biosphere model. Please, elaborate on this in more detail and in relation to the anticipated forest ecosystem approach.

The term "damage" is frequently used in the ms. in different contexts and scales: "Damage of photosynthetic apparatus", "ozone damage", "leaf-scale" to "global estimates". In some parts it seems that damage functions refer to "the effects of ozone uptake on photosynthetic variables" and in other parts damage seems to refer to "the fractional loss of carbon uptake associated with ozone uptake". We suggest that the authors define explicitly and very early in the ms. what they mean with "damage", "ozone damage", "damage function" and also specify the difference between "dose-response relationship" and "damage functions".

Ozone and trees and forests: The actual extent to which reduction in tree growth due to ozone occurs in the real forest remains still unclear. There are several studies which found significant effects, others did not (they did not observe measured above-ground tree growth, which is not total biomass, but an often used proxy for it). We think these controversial results should be considered and discussed as they may help to better contextualize the paper.

Juvenile vs. mature trees: Despite the short explanation given on p. 8, line 21-28, it is not clear how DDRs for mature trees were simulated. Since this is a very important step (and output) for the non-modelers, a more detailed explanation will be very useful here.

Reduction of biomass: It is not clear what is intended here as reduction of biomass. While we understand the reduction of biomass increment, we can hardly see a living tree reducing its biomass due to ozone. The formulation (6) actually seems to refer to a difference of biomass of treated trees with respect to the controlled ones, and not to a reduction of biomass of the treated trees. This is somewhat acknowledged by the authors in the discussion, but perhaps it deserves more emphasis.

Finally, it will be important to have some statement how the authors - based on these results - see the value of the risk maps produced by EMEP for e.g. European forests.

Specific comments:

P2, L17: "simulated reductions in GPP due to ozone damage vary substantially between models and model versions": please, provide some examples and values.

P2, L18: Here and elsewhere, you may consider Cailleret et al. (2018) as additional reference.

P3, L10: You may consider Schaub et al. (2005) as additional reference. M&M: Please, provide more details on O-CN structure and main assumptions, even though this model has been used and described in Zaehle and Friend (2010), and in Franz et al. (2017). It would help that the reader does not need to go back and forth between the current paper and these ones. e.g., What is the spatial resolution? Individual-based or cohort-based model?

P4, L31-33: No reserves?

P4, L31-33: Biomass growth seems to be dependent only on source but not on sink activity. Is that correct? If yes, this is a strong assumption and limit of the modelling approach (see Körner 2015) that has to be discussed later.

P5, L10: We suggest to add the equation(s) used to calculate An,l

P5, L17: It seems that the authors assume that O3 concentration is constant within the canopy. Correct? Please, clarify and discuss.

P5, L25: "the Phytotoxic Ozone Dose (POD, mmolm−2) can be diagnosed by the accumulation of fst,l for the top canopy layer (l = 1)." In most ozone-flux modeling approaches, POD is calculated based on a "big-leaf" approach (one layer of leaf area, but LAI can be > 1; approach used in DO3SE) -> this is different from the accumulation of fst,l for the top canopy layer. See also P16, L18. Please clarify.

P6, L18: Correct "ration"

P6, L19: Correct "is is"

P7, L18: The initialization phase is not clear: we don't see how the model can run "from

bare ground until the simulated stand-scale tree age was stable and representative of 1-2 year old seedlings". And this is even less clear with the sentence P8, L5: "The duration of the initialization phase (...) averages 7.8 years". Furthermore, did the authors run only one or multiple O-CN simulations per study case (per experiment)? We guess there are some stochastic processes in O-CN, these ones may induce some epistemic uncertainties that have to be considered in the modeling framework.

P9, L25: It is not clear how this "tuning" has been performed: was it a manual or an automatic optimization. Which algorhythm was used? Bayesian framework? Which metric did the authors try to optimize (likelihood; rmse, r2)?

P10, L15: Please, show results in Supp Mat.

P11, L2: "The simulations L12PS and L12VC (...) strongly overestimate". Yes, but this is less strong than the underestimation by W07 and L13.

Figure 2, panels a, b, c, d: what do the simulations without O3 fumigation (after the red line) look like?

Figure 2: Please, add simulated before cumulative in the legend; Idem P11, L11, add simulated before CUOY.

P11, L16: There is no control simulation shown in Fig. 2 (see our comment above)

P12, L5-7: In M&M

P12, L12 to P13, L4: This comparison between mature vs. young trees is not described in the M&M. How do the simulations differ in terms of initialization etc.?

P14, L11 and P15, L8: Please provide some values.

P15, L17 and throughout the paper: Note that Büker et al. (2015) used the Jarvis equation to simulate stomatal conductance while the Ball & Berry one is used here. Please be cautious when comparing both studies.

P15, L20-22: We agree that this is a key aspect, which has to be more detailed. Discussion: The DRRs built in the present study are valid only for O-CN and may not work for other dynamic vegetation models (strongly depends on how biomass growth is simulated by the DVM -> sink vs. source activity etc.). This is implicitly written in P15, L32-33; but this has to be mentioned again in the conclusion. We suggest to rather highlight that the approach developed here is interesting and can be followed to calibrate "ozone submodels" in further DVMs.

P16, L15 and P17, L5: Idem show some results in Supp Mat.

P17, L30-33: Authors ask for monitoring programs "capable to measure the actual increment of biomass". We assume that they know that these programs do exist, e.g. national forest inventories and international monitoring programs such as the ICP Forests. Please, quote these programs here.

P18, L1-10: The authors may also consider that trees usually occur in forests, and that forests are subjected to entire ecosystem dynamics that can offset / mitigate / adapt / compensate ozone effects. This should be discussed and considered in the conclusions.

Suggested references:

Cailleret et al. (2018) Ozone effects on European forest growth - towards an integrative approach. Journal of Ecology, doi: 10.1111/1365-2745.12941

Novak et al. (2008) Ozone effects on visible foliar injury and growth of Fagus sylvatica and Viburnum lantana seedlings grown in monoculture or in mixture. Environmental and Experimental Botany, doi: 10.1016/j.envexpbot.2007.08.008

Schaub et al. (2005) Physiological and foliar symptom response in the crowns of Prunus serotina, Fraxinus americana, and Acer rubrum canopy trees to ambient ozone under forest conditions. Environmental Pollution, doi: 10.1016/j.envpol.2004.06.012

---

## Author Response (AR1)

Dear Editor,

Many thanks for your reply. Please find our answers to your comments, the answers to the referees (indicating major changes in the new manuscript) and a marked up manuscript version showing the conducted changes below.

Best wishes

Martina

**0.1 Answers to Editor**

Q: Please give particular attention to ensuring that you more fully addressed the comments of reviewer # 1 with respect to the top of canopy dosage and variation with depth through the canopy. Whilst I found the authors responses satisfactory, I wonder how much effort it would be to test an alternative assumption? If only to demonstrate the sensitivity to an alternative assumption? Alternatively, some space could be dedicated to this in the discussion.

A: As mentioned by referee # 1 the estimate of the PODy values strongly impacts the resulting dose-response-relationships. The PODy values in the Büker et al. (2015) study are modelled values and not measured ones. Büker et al. (2015) calculate the PODy according to the LRTAP-Convention (2010). To be able to compare our simulation results to their ones we have to use the same approach as they do. As already discussed in the manuscript differing model features between $DO_3SE$ and O-CN will probably impact estimates of the $PODy$ and hence the dose-response-relationships.

The simulation of ozone uptake and damage through all canopy layers is unconnected to the calculation of the $PODy$ used for the formation of the biomass dose-response-relationships and thus open to the application of different approaches. We tested 2 commonly used approaches to simulate ozone uptake and damage within the canopy. First the explicit simulation of ozone uptake in each canopy layer and the application of the respective damage fraction in each layer (damage based on $CUOY$). Ozone uptake in each layer is determined by factors that impact the stomatal conductance in each canopy layer, e.g. the light conditions and the varying nitrogen content within the crown. Secondly we tested the calculation of the ozone uptake only in the top canopy layer of the trees and the application of the respective damage fraction to all canopy layers (damage based on $PODy$) and elaborate on the

differences between the 2 and the reasons for them.

If further data become available regarding the vertical gradient of ozone uptake within canopies alternative simulation approaches might be developed. The sentence on p.16 ll.20-23 was extended to make this point: 'More analysis of the differential effect of ozone injury within deep canopies are required to evaluate whether the scaling of top-of-the-canopy injury to whole canopy injury is appropriate or if alternative simulation approaches need to be developed (now on p.17 ll.10-12).'

Q: In addition, please pay particular to reviewer # 2, where they ask about mature vs young trees and species-specific relationships. These questions strike me as interesting points to form a discussion around future extensions of your work.

A: Both aspects (young vs. mature trees and species-specific relationships) are now discussed in 2 paragraphs ((now on p.17 l.23-p.18 l.2) and p.19 ll.5-15) of the discussion.

In addition to their comments, could I please ask you to:

Q: Abstract: define $V_{cmax}$

A: Done.

Q: Pg 4: where you describe how the labile non-structural pool buffers growth, it strikes me as important for the reader to get a sense of what sort of time-scale this may impose on any direct impact of ozone on leaf-photosynthesis vs. realised growth. Days? Weeks? Months?

A: We included a sentence to explain that the labile pool responds within days to changes in GPP, and the long-term reserve takes several months to respond (p.4 ll.33-p.5 l.2).

Q: Pg 5: Where you refer to the CUOY being calculated by summation over all layers and then refer to Franz et al. 2017. Is there further detail that the reader should see here? If it is simply summation, then perhaps simply cite Franz? As written it implies that there is additional insight here and that should either be presented here or clarified.

A: We changed the manuscript from (see Franz et al. 2017 for details) to (Franz et al. 2017).

Q: Pg 6, line 25: why didn't they match well? Could this point be developed/shown? My reading of the manuscript is that this is this what is referred to in Figure 1? If so, could the authors explain to the reader that they will address this in the results? My reading of the methods is that it is simply asserted to be true, but I may have missed the explanation...

A: The respective sentence is a forecast of the results displayed in Figure 1. As suggested we added a remark that the corresponding results are shown in the results section and Fig. 1 (p.7 ll.10-11).

Q: Equation 6, what is n?

A: The 'n' is a typing error and got removed.

Q: Fig 2: CUOY not CUOy.

A: Changed.

Q: Page 12, line 10: This statement is true so long as the canopy is simulated to be estimated via the $V_{cmax}$ limitation? This would change in different scenarios (i.e. if the canopy was $J_{max}$ limited) and I think this point should be clarified for the reader. Currently it (wrongly - in my opinion) implies that a model would get the same result if they applied the function to $V_{cmax}$ or An.

A: Please note that not only $V_{cmax}$ but also $J_{max}$ is changed (see Methods), hence the ratio between $V_{max}$ and $J_{max}$ remains constant. We added '$V_{cmax}$ and simultaneously $J_{max}$' in the respective sentence to remind the reader of this fact (p.12 l.11).

**0.2 Answers to Anonymous Referee # 1**

Q: The authors assume that the modelled accumulation of ozone fluxes at the top canopy layer equals POD during the model-observation comparison process. Please justify this assumption. I think this is important for the evaluation of model against observation, considering the ozone damage is explicitly calculated through the canopy and integrated to derive the whole tree damage. The modelled POD value largely influences the slope of the

resultant dose-response curve and its distance with observed dose-response curve. I am wondering how would the authors account for this treatment in influencing the evaluation of different algorithms against observed data.

A: We designed our study such that our way to calculate $POD$ is consistent with those from Büker et al. (2015), from which we took the dose-response-relationships. They calculate the $PODy$ used for their analysis in accordance with the LRTAP-Convention (2010), which states 'the index PODY is used to quantify the flux of ozone through the stomata of the uppermost leaf level that is directly exposed to solar radiation and thus no calculation of light exclusion, caused by the filtering of light through the leaves of the canopy, is required'. We calculate the $PODy$ based on the LRTAP-Convention (2010), to be able to compare our simulation results to those of Büker et al. (2015).

We will add a citation of the LRTAP convention to the explanation on the calculation of POD in the text: 'For comparison to observations, the Phyto-toxic Ozone Dose ($POD$, $mmol\,m^{-2}$) can be diagnosed by the accumulation of $f_{st,l}$ for the top canopy layer (l = 1), in accordance with LRTAP-Convention (2010) and Büker et al. (2015). (p.6 l.11)'

Of course, there will be uncertainty in the calculation of the $POD$ by both Büker et al. (2015) and our study compared to the real-world $POD$, given both are based on different, but evaluated models (Emberson et al., 2000; Franz et al., 2017), but in the absence of direct measurements of $POD$ it is impossible to judge whether or not this would introduce any systematic bias into the comparison.

Q: I am curious why did not the author try to use different damage functions at different depth of the canopy?

A: Each of the damage functions is applied to all canopy layers in separate simulations for each damage function. The ozone damage differs within the canopy, as increasing canopy depth leads to lower leaf-specific photosynthesis, conductance, and therefore ozone uptake and damage.

Our aim was to investigate the suitability of different damage functions to reproduce observed biomass damage relationships. Following this we always only applied one damage function in one simulation. The application of different damage functions in one simulation, e.g. different damage functions for different canopy layers, can not contribute to answer our research question.

Evidence exists that sunlit and shades leaves exhibit a different sensitivity to $O_3$ (Tjoelker et al., 1995; Wieser et al. 2002). Following this the application

of different damage functions for different canopy layers might yield improved damage estimates. However damage relationships for different canopy depth are to our knowledge not available as well as independent data to evaluate them.

Q: 3) Another important, but still largely missing, aspect in simulating ozone impacts on vegetation is the huge diversity of species-sensitivity in an ecosystem. Dealing with vegetation to the PFT level is not enough, though totally make sense in terms of large scale modelling and data scarcity. This work could be improved by further talking about diversity of species response to ozone. To this end, I found the following work could be a good reference: Wang, B. et al. Forests and ozone: productivity, carbon storage, and feedbacks. Sci. Rep. 6, 22133; doi: 10.1038/srep22133 (2016)

This study, though without sophisticated ozone damage simulation, had an explicit simulation of species sensitivity to ozone using an individual-based model and found dampened responses to ozone over long-term simulations.

A: We will include this study in our discussion (p.17 ll.27-29).

**0.2.1   Minor comments**

Q: L27 on page 4: please justify the statement of highest N concentration at the top of the canopy and its exponential decline with increasing canopy depth.
A: We base this statement on the publications by Friend (2001) and Niinemets et al. (2015) and will add these references in the manuscript (p.4 ll.27-29).

Q: L18 on page 5: in equation 2, how is the stomatal conductance of O3 calculated?
A: Explanation added (p.5 ll.25-27).

Q: L28 on page 8: identical →identically.
A: Done.

Q: L5-6 on Page 18: this sentence should be restructured to make it easier to follow.

A: Done.

Q: L7 on page 18: ''all in all' should be followed a comma.
A: Done.

**0.3 Answers to Referee Marcus Schaub and comments by Maxime Cailleret and Marco Ferretti**

Q: The authors argue that so far applied damage functions result in impacts with large uncertainty in the magnitude of ozone effects predicted. They use the O-CN biosphere model to test four already existing damage functions in terms of their simulated whole-tree biomass responses against field data from 23 ozone filtration/fumigation experiments and found that biomass damage was overestimated (Lombardozzi et al. 2012) or underestimated (Wittig et al 2007; Lambardozzi et al. 2013). The authors tune/reparameterize those damage functions towards a better fit with data from 15 fumigation experiments with young trees. In a second step, the authors tune DRRs again so that relative biomass (or NPP) simulated on adult trees fit the measured values on young trees.

A: As we explain below, this is an accurate representation of the manuscript's content, with the exception of the last sentence, as we did not recalibrate the model for adult trees.

Q: The ms. reads very well and is certainly within the scope of BG. As a matter of fact, we appreciate this exercise as it addresses a crucial issue in ozone risk assessment and provides an excellent review on the state of the art.

A: Thank you.

Q: While the first part, i.e. recalibration of existing damage functions makes sense to improve DRRs for young trees and better predict biomass loss due to ozone. We are, however, concerned about the second step, i.e. the reparameterization/tuning of those functions (for young trees) to better predict

relative biomass for mature trees.

A: It is a misunderstanding that we re-calibrated the model to simulate old-growth forests. This is not the case. We applied the model tuned for young trees to simulate old growth forests and compared the simulated ozone damage of the young and mature trees in terms on their effect of biomass and biomass production. We found that young and mature tree produce strongly differing biomass dose-response-relationships but similar dose-response- relationships for NPP. This leads us to the assumption that NPP responses to ozone damage of young and mature trees might be better comparable than biomass responses. We believe that this analysis is helpful to illustrate the problem of using biomass-reduction-ozone uptake relationships to quantify ozone damage for the development of process-based models. We will rework the manuscript in order to make sure the confusion regarding the re-calibration cannot occur. We introduced a subsection called ' Modelling protocol for mature trees' and we state in there 'The ozone injury for mature trees is calculated based on the same $tun_{VC}$ injury function (see Tab. **??**) that is used in the simulation of young trees.'(p.9 ll.11-12)

Q: The authors aim at improving the quantitative understanding of ozone effects on forest growth and carbon sequestration on a regional or even global scale. Using data from seedlings gown under (semi-)controlled experiments ranging over a few years may (still) not lead to reliable model functions for adult trees growing in complex forest ecosystems. The cited work by Franz et al. 2017 (GPP reduction, based on damage functions from Wittig et al. 2007) is an example how model exercises using modeled data may result in inaccurate predictions – if not validated with measured data from adult trees (see also Cailleret et al. 2018). Page 15, line 6-10 demonstrates the risk of applying models, based on former functions and stresses the need of validating model exercises with measured data (e.g. from ICP Forests).

A: The model experiment with adult trees presented were an attempt illustrate this problem and the question of how much these damage functions may be scalable or not. We will add the manuscript by Cailleret et al. 2018 to the discussion, but remain sceptical that comparison to ICP forest data, given the lack of a control to isolate ozone damage from other co-occurring environmental drivers such as atmospheric N and S deposition or climatic variability. Nevertheless, this is a good suggestion to corroborate not the damage function per-se, but at least the simulated growth rates under O3 exposure. We will consider this as a follow-up study, subject to sufficient data availability to run and evaluate models for these sites.

Q: We suggest to either omit the second part or to extend section 3.3. and the discussion and to outline not only the advances but also the still existing lack of knowledge for estimating ozone induced biomass effects on adult trees, forest ecosystems respectively.

A: We will extend our discussion to highlight this uncertainty.(p.19 ll.9-13)

Q: Novak et al. (2008) found that species competition may alter DDRs. We did not understand if and how competition is considered in the O-CN biosphere model. Please, elaborate on this in more detail and in relation to the anticipated forest ecosystem approach.

A: As described, (P 7, LL 13-15 of the old manuscript) OCN simulates plant functional types, not species. Therefore the effects of species composition and its change are not accounted for. This issue is taken up in the discussion (p.17 l.29 - p.18 l.2).

Q: The term "damage" is frequently used in the ms. in different contexts and scales: "Damage of photosynthetic apparatus", "ozone damage", "leaf-scale" to "global estimates". In some parts it seems that damage functions refer to "the effects of ozone uptake on photosynthetic variables" and in other parts damage seems to refer to "the fractional loss of carbon uptake associated with ozone uptake". We suggest that the authors define explicitly and very early in the ms. what they mean with "damage", "ozone damage", "damage function" and also specify the difference between "dose-response relationship" and "damage functions".

A: We will revise the manuscript according to these suggestions. We define the use of the respective terms on p.5 ll.8-11.

Q: Ozone and trees and forests: The actual extent to which reduction in tree growth due to ozone occurs in the real forest remains still unclear. There are several studies which found significant effects, others did not (they did not observe measured above-ground tree growth, which is not total biomass, but an often used proxy for it). We think these controversial results should be considered and discussed as they may help to better contextualize the paper.

A: We will include this in the discussion.(p. 19 ll.7-15 )

Q: Juvenile vs. mature trees: Despite the short explanation given on p. 8, line 21-28, it is not clear how DDRs for mature trees were simulated. Since this is a very important step (and output) for the non-modelers, a more detailed explanation will be very useful here.

A: As stated above, we did not re-tune the relationships.

Q: Reduction of biomass: It is not clear what is intended here as reduction of biomass. While we understand the reduction of biomass increment, we can hardly see a living tree reducing its biomass due to ozone. The formulation (6) actually seems to refer to a difference of biomass of treated trees with respect to the controlled ones, and not to a reduction of biomass of the treated trees. This is somewhat acknowledged by the authors in the discussion, but perhaps it deserves more emphasis.

A: We will clarify the text to be more precise that we mean the difference between a control and a treatment. (p.9 l.14)

Q: Finally, it will be important to have some statement how the authors - based on these results - see the value of the risk maps produced by EMEP for e.g. European forests.

A: It is not the point of the paper to provide an assessment of EMEP and its suitability for risk assessment, specifically because we only assess one part, the O3-damage calculation, of the EMEP risk assessment, while EMEP includes a range of other processes. A proper assessment of these Maps would require a simulation comparable to the EMEP projections to disentangle regional and temporal differences in O3-related risks. Such wall-to-wall simulations are beyond the scope of our study, and we would therefore prefer to avoid to add too much speculative assessment at this point.

**0.3.1 Specific comments**

Q: P2, L17: "simulated reductions in GPP due to ozone damage vary substantially between models and model versions": please, provide some examples and values.

A: We added suitable references. The modelling protocols, in these studies differ strongly regarding the simulated years and accounting/not accounting for e.g. changing nitrogen deposition and elevated $CO_2$. For instance Sitch

et al. (2007) estimates the GPP reduction between 1901-2100 and Franz et al. (2017) the mean decadal reduction for 2001-2010 compared to a simulation without accounting for ozone effects. Lombardozzi et al. (2012) report mean annual reductions in GPP for a 20 yr run of CLM at 100 ppm $O_3$ which can be compared to neither of the first. A concise summary of the estimated GPP reductions in these studies without explaining the different modeling assumptions might mislead the reader. We believe that a detailed display of the results and the underlying assumptions would to much extend the introduction.

Q: P2, L18: Here and elsewhere, you may consider Cailleret et al. (2018) as additional reference.

A: We will include the Cailleret et al. paper.

Q: P3, L10: You may consider Schaub et al. (2005) as additional reference. M&M: Please, provide more details on O-CN structure and main assumptions, even though this model has been used and described in Zaehle and Friend (2010), and in Franz et al. (2017). It would help that the reader does not need to go back and forth between the current paper and these ones. e.g., What is the spatial resolution? Individual-based or cohort-based model?

A: We will add a short description of relevant model features, but refrain from repeating what has been described before. Furthermore in the simulations here the model is run on point scale (the coordinate of the experiment site) and no spatial resolution is applied. (p.4 l.21 and p.8 l.2)

Q: P4, L31-33: No reserves?

A: See description of storage on P4, L29-31 (p.4 32-33 in new manuscript).

Q: P4, L31-33: Biomass growth seems to be dependent only on source but not on sink activity. Is that correct? If yes, this is a strong assumption and limit of the modelling approach (see Körner 2015) that has to be discussed later.

A: This is not fully true, as the model does account for sink limitation due to nutrient constraints. It is correct, however, that the model does not account for sink limitation due to constrained rooting zone volume or number of leaf buds.

Q: P5, L10: We suggest to add the equation(s) used to calculate An,l

A: Given the complexity of the approach, as outlined in Kull & Kruit 1998, we prefer to not do this, as it would divert the reader's attention. We will add the variables that drive the calculations of $A_{n,l}$, namely leaf internal partial pressure of $CO_2$, absorbed photosynthetic photon flux density on shaded and sunlit leaves, leaf temperature, as well as the maximum carboxylation and electron-transport rates, which each are a function of leaf nitrogen concentration.

Q: P5, L17: It seems that the authors assume that O3 concentration is constant within the canopy. Correct? Please, clarify and discuss.

A: Yes, the ozone concentration is constant within the canopy. The biomass damage experiments we try to reproduce here are all conducted with saplings which were mostly fumigated in open top chambers. Pronounced $O_3$ gradients within the canopy thus are not to be expected. However, we note that ozone uptake is not constant within the canopy given the distribution of light and photosynthetic capacity. We assume that these gradients have a much stronger effect on layered O3 uptake than any vertical gradient in O3 in such an experimental setting.

Q: P5, L25: "the Phytotoxic Ozone Dose (POD, mmolm-2) can be diagnosed by the accumulation of fst,l for the top canopy layer (l = 1)." In most ozone-flux modeling approaches, POD is calculated based on a "big-leaf" approach (one layer of leaf area, but LAI can be ¿ 1; approach used in DO3SE) -¿ this is different from the accumulation of fst,l for the top canopy layer. See also P16, L18. Please clarify.

A: As in our reply to referee #1:

We designed our study such that our way to calculate $POD$ is consistent with those from Büker et al. (2015), from which we took the dose-response-relationships. They calculate the $PODy$ used for their analysis in accordance with the LRTAP-Convention (2010), which states 'the index PODY is used to quantify the flux of ozone through the stomata of the uppermost leaf level that is directly exposed to solar radiation and thus no calculation of light exclusion, caused by the filtering of light through the leaves of the canopy, is required'. We calculate the $PODy$ based on the LRTAP-Convention (2010), to be able to compare our simulation results to those of Büker et al. (2015).

We will add a citation of the LRTAP convention to the explanation on the

calculation of POD in the text: 'For comparison to observations, the Phytotoxic Ozone Dose ($POD$, $mmol m^{-2}$) can be diagnosed by the accumulation of $f_{st,l}$ for the top canopy layer (l = 1), in accordance with LRTAP-Convention (2010) and Büker et al. (2015).'

Q: P6, L18: Correct "ration"

A: Done.

Q: P6, L19: Correct "is is"

A: Done.

Q: P7, L18: The initialization phase is not clear: we don't see how the model can run "from bare ground until the simulated stand-scale tree age was stable and representative of 1-2 year old seedlings". And this is even less clear with the sentence

A: Our aim is to simulate seedling similar to the fumigated seedling in the biomass damage experiments. When we first start our model no trees are present and seedlings start to grow after the simulation starts. O-CN is a stand-scale model and not an individual based model. Until the mean stand-scale age of 1-2 years is realised a larger number of simulation years passes. The exact number of simulation years is site specific but on average over all simulation sites it takes the mentioned 7.8 years. After this initialisation phase we can start the simulation of the experiment years.

Q: P8, L5: "The duration of the initialization phase (. . .) averages 7.8 years". Furthermore, did the authors run only one or multiple O-CN simulations per study case (per experiment)? We guess there are some stochastic processes in O-CN, these ones may induce some epistemic uncertainties that have to be considered in the modeling framework.

A: There are no stochastic elements in these simulations and running multiple simulations yields identical results.

Q: P9, L25: It is not clear how this "tuning" has been performed: was it a manual or an automatic optimization. Which algorhythm was used? Bayesian framework? Which metric did the authors try to optimize (likelihood; rmse, r2)?

A: The tuning was performed manually (p.7 l.11).

Q: P10, L15: Please, show results in Supp Mat.

A: As stated in the manuscript accounting for direct injury of the stomates had only minimal effects (and only for the needleleaf category), see also Fig. 1 here. We could include the graphic into the Supp. Material but we are not sure if it justifies the creation of a supplement since there is none at the moment.

Q: P11, L2: "The simulations L12PS and L12VC (. . .) strongly overestimate". Yes, but this is less strong than the underestimation by W07 and L13.

A: We do mention that W07 strongly underestimates the damage and elaborate on the results of L13 and the reasons for them (P10,LL1-14 in the manuscript in discussion).

Q: Figure 2, panels a, b, c, d: what do the simulations without O3 fumigation (after the red line) look like?

A: In the control simulation no ozone damage occurs, photosynthesis and biomass do not decline.

Q: Figure 2: Please, add simulated before cumulative in the legend; Idem P11, L11, add simulated before CUOY.

A: Done.

Q: P11, L16: There is no control simulation shown in Fig. 2 (see our comment above)

A: Yes, because this graphic is used to explain the extreme effects of ozone fumigation on the plant physiological processes when applying the $L12_{PS}$ injury relationship. In the control simulation no ozone damage occurs, photosynthesis and biomass do not decline. In our view adding the control here overcrowds the graph but does not add valuable information to the reader. The key point of this graphic is to illustrate the extreme effects ozone fumigation imposes on plant performance (e.g. negative values of $A_n^{can}$) if the $L12_{PS}$ injury relationship is applied.

Q: P12, L5-7: In M&M P12, L12 to P13, L4: This comparison between mature vs. young trees is not described in the M&M. How do the simulations differ in terms of initialization etc.?

A: See P8 L21-30 of the original manuscript, which is part of the M&M. We will make sure that this text is not overlooked in the revised version by introducing a separate subsection called ' Modelling protocol for mature trees' (p.9 l.3)

Q: P14, L11 and P15, L8: Please provide some values.

A: This is impossible without performing a large-scale integration of the model, which is beyond the scope of this paper.

Q: P15, L17 and throughout the paper: Note that Büker et al. (2015) used the Jarvis equation to simulate stomatal conductance while the Ball & Berry one is used here. Please be cautious when comparing both studies.

A: It is unclear to us, why the form of the equation is relevant here. What is relevant is whether the simulated canopy conductance is in agreement with observations, and at least for the part of OCN this has been demonstrated by Franz et al. 2017.

Q: P15, L20-22: We agree that this is a key aspect, which has to be more detailed.

A: Both model differ in many respects that will impact the estimate of ozone uptake and accumulation as stated in the manuscript and the provided examples. We are not sure if adding a longer, more detailed list of the differing model features will benefit the reader. However we extended the respective sentence to clarify that these impact the suggested dose-response-relationships: 'However, both models vary in their complexity of the simulated plants, carbon assimilation, and growth processes, which will also impact the estimates of ozone accumulation ($PODy$) and hence their suggested biomass dose-response-relationships.'(p.16 ll.8-10)

Q: Discussion: The DRRs built in the present study are valid only for O-CN and may not work for other dynamic vegetation models (strongly depends on how biomass growth is simulated by the DVM -¿ sink vs. source activity etc.). This is implicitly written in P15, L32-33; but this has to be mentioned again in the conclusion. We suggest to rather highlight that the approach developed

here is interesting and can be followed to calibrate "ozone submodels" in further DVMs.

A: Done (p.20 ll.2-5).

Q: P16, L15 and P17, L5: Idem show some results in Supp Mat.

A: Regarding P16, L15: As stated in the manuscript the calculation of plant injury based on $POD1$ rather than $CUO1$ (using an adapted slope in the model simulations) yielded dose-response-relationships which are comparable to the ones based on $CUO1$, see also Fig. 2 here. The simulation of plant injury based on $CUO1$ seems to be preferable over $POD1$, because the canopy layer specific ozone uptake is translated into a layer specific injury fraction. Following this we remain uncertain as to the value of including the $POD1$ results into the supplement. Regarding P17, L5: See answer to Q P10, L15.

Q: P17, L30-33: Authors ask for monitoring programs "capable to measure the actual increment of biomass". We assume that they know that these programs do exist, e.g. national forest inventories and international monitoring programs such as the ICP Forests. Please, quote these programs here.

A: We will clarify that we meant to state monitoring programs of ozone damage. Of course, increment networks exists, but they are intrinsically incapable of separating the effects of ozone, N and S deposition and climate variations.

Q: P18, L1-10: The authors may also consider that trees usually occur in forests, and that forests are subjected to entire ecosystem dynamics that can offset / mitigate / adapt / compensate ozone effects. This should be discussed and considered in the conclusions.

A: Taken up in the discussion (p.17 ll.29 - p.18 l.2 and p.19 ll.7-15).

**0.4   References**

[revised manuscript text omitted]

---

## Referee Report (RR1)

Review Biogeosciences Discussion of

Franz et al.: **Evaluation of simulated biomass damage in forest ecosystems induced by ozone against observation-based estimates**

by Marcus Schaub and Marco Ferretti

We acknowledge the author's efforts of adding clarifications to the manuscript and considering our comments. The quality of the manuscript has considerably improved and – in our opinion – is ready to be published after addressing the following points:

(i) Title: The title "Evaluation of simulated biomass damage in forest ecosystems induced by ozone against observation-based estimates" is true when only the part related to the experiments with young trees is concerned. Mature trees are simulated, not actually observed. We therefore suggest to change the title accordingly. We consider this as an important point as the title is not a minor element and will influence the perception and how the manuscript will be received.

(ii) Young *vs*. mature trees: While it is perfectly justified to show the impact of the different response functions on the possible outcomes of seedlings experiment, extrapolation to forest ecosystem remains a strong simplification. We suggest that authors should clearly state that they use dose-response relationship derived from experiments with young trees for their evaluations regarding mature trees, assuming such relationships hold valid, but that they are well aware this may not be the case. In their new discussion paragraph (p. 19, ll. 9-13) they may also suggest the possible consequences of this assumption.

(iii) It is true that the manuscript deals with only one part of the entire EMEP approach. Nevertheless, the risk evaluation relies very much on the dose-response functions. We still believe that the authors should comment on possible, related consequences of their findings in this respect.

(iv) Throughout the manuscript, the authors used the terms "young trees", "cuttings", "saplings", "seedlings" and "small trees" to describe the experimental trees. We suggest to bring in some consistency by using only e.g. young trees.

Specific comments:

Page 1, line 10: delete "field" in "…against field data …" as this is misleading. Data are from experiments not from field observations.

Page 1, line 11: add "… experiments conducted with young trees from European trees species …"

Page 1, line 12: delete "simulated" in "… functions lead to simulated whole-tree …" as this is redundant; it is clear that functions lead to simulated results.

Page 3, line 3: replace small trees by young trees (see comment iv)

Page 8, line 17: cuttings? (see comment iv)

Page 19, line 27: correct "Whether the simulation of injury … can indeed be transferred to adult trees or not to yield realistic …"

---

## Author Response (AR2)

Dear Editor,

please find our answers to the referees Marcus Schaub and Marco Ferretti and a marked up manuscript version showing the conducted changes below.

Best wishes

Martina

**1 Answers to Referee Marcus Schaub and Marco Ferretti**

We acknowledge the author's efforts of adding clarifications to the manuscript and considering our comments. The quality of the manuscript has considerably improved and - in our opinion - is ready to be published after addressing the following points:

Q: Title: The title "Evaluation of simulated biomass damage in forest ecosystems induced by ozone against observation-based estimates" is true when only the part related to the experiments with young trees is concerned. Mature trees are simulated, not actually observed. We therefore suggest to change the title accordingly. We consider this as an important point as the title is not a minor element and will influence the perception and how the manuscript will be received.

A: The title was rephrased to: "Evaluation of simulated ozone effects in forest ecosystems against biomass damage estimates from fumigation experiments"

Q: Young vs. mature trees: While it is perfectly justified to show the impact of the different response functions on the possible outcomes of seedlings experiment, extrapolation to forest ecosystem remains a strong simplification. We suggest that authors should clearly state that they use dose-response relationship derived from experiments with young trees for their evaluations regarding mature trees, assuming such relationships hold valid, but that they are well aware this may not be the case. In their new discussion paragraph (p. 19, ll. 9-13) they may also suggest the possible consequences of this assumption.

A: Already in the introduction (p. 3 ll. 11-14) we introduce the uncertainty of transferring results obtained from young trees to mature tree: 'However, the

effects of ozone on leaf physiology (e.g. net photosynthesis and stomatal conductance) or plant carbon allocation may differ between juvenile and adult trees (Hanson et al., 1994; Samuelson and Kelly, 1996; Kolb and Matyssek, 2001; Paoletti et al., 2010). Whether or not biomass dose-response relationships can be used to calibrate injury functions for mature trees is uncertain.'.

In the subsection 'Modelling protocol for mature trees' we clearly state that the damage relationship developed for young trees is applied in the simulation of mature trees: 'The ozone injury for mature trees is calculated based on the same $\text{tun}_{VC}$ injury function (see Tab. 1) that is used in the simulation of young trees.'(p. 9 ll. 11-12). We explain that this $\text{tun}_{VC}$ injury function is derived by tuning our simulation results against the dose-response-relationships by Büker et al. (2015) (p. 10 ll. 14-15). We state repeatedly that the data set by Büker et al. (2015) bases on experiments with young trees p. 3 ll. 32-34 and p. 8, l. 13. In a further step we evaluate if the simulation of mature trees yields comparable dose-response-relationships to young trees, as both apply identical injury functions. We find this is not the case in our simulations and clearly state in the discussion: e.g. 'If the simulation of injury to photosynthesis based on experiments with young trees can indeed be transferred to adult trees to yield realistic biomass damage estimates is still uncertain.'(p. 19 ll. 7-9) and conclusion: 'The comparison of simulated biomass dose-response relationships of young and mature trees shows strongly different slopes. This suggests that observed biomass damage relationships from young trees might not be suitable to estimate biomass damage of mature trees.'(p. 20 ll. 5-7).

We extended the sentence in the subsection 'Modelling protocol for mature trees'(p. 9 ll. 11-12) to link details on the development of the $\text{tun}_{VC}$ injury function as this is explained only in the below section: 'The ozone injury for mature trees is calculated based on the same $\text{tun}_{VC}$ injury function (see Tab. 1) that is used in the simulation of young trees (see subsection 2.5 for details on the development of $\text{tun}_{VC}$).'

Q: It is true that the manuscript deals with only one part of the entire EMEP approach. Nevertheless, the risk evaluation relies very much on the dose-response functions. We still believe that the authors should comment on possible, related consequences of their findings in this respect.

A: We added: 'Our results highlight the importance for improved evaluation of injury functions applied in the simulation of ozone damage for large-scale risk assessments, and we discuss a number of important considerations for an improved parameterisation below.' to the discussion section (p. 15 ll. 6-8

in the new manuscript version) and extended a sentence in the conclusion: 'Injury functions included into terrestrial biosphere models are a key aspect in the simulation of ozone damage and have a great impact on the estimated damage in large-scale ozone risk assessments.' (p. 19 l. 33 - p. 20 l. 1) to take up this point again.

Q: Throughout the manuscript, the authors used the terms "young trees", "cuttings", "saplings", "seedlings" and "small trees" to describe the experimental trees. We suggest to bring in some consistency by using only e.g. young trees.

A: We exchanged 'seedlings', 'saplings', 'small trees' by young trees. The term 'cuttings' is used only once (p. 8 l. 13: 'The field experiments were conducted on young trees or cuttings.') to describe the trees that went into the Büker et al. (2015) study. We believe mentioning that part of the trees were cuttings adds additional information on the properties of trees in the experiments and we believe it is sensible to keep it.

**1.1 Specific comments:**

Q: Page 1, line 10: delete "field" in "...against field data ..." as this is misleading. Data are from experiments not from field observations. A: Deleted.

Q: Page 1, line 11: add "... experiments conducted with young trees from European trees species ..." A: Done.

Q: Page 1, line 12: delete "simulated" in "... functions lead to simulated whole-tree ..." as this is redundant; it is clear that functions lead to simulated results. A: We don't think this is the case, and would like to keep it. We had other comments in directly the opposite direction on various occasions.

Q: Page 3, line 3: replace small trees by young trees (see comment iv) A: Done.

Q: Page 8, line 17: cuttings? (see comment iv) A: As stated above: The term 'cuttings' is used only once to describe the trees that went into the Büker et

al. (2015) study. We believe mentioning that part of the trees were cuttings adds additional information on the properties of trees in the experiments and we believe it is sensible to keep it.

Q: Page 19, line 27: correct "Whether the simulation of injury ... can indeed be transferred to adult trees or not to yield realistic ..." A: We believe that the sentence: 'If the simulation of injury to photosynthesis based on experiments with young trees can indeed be transferred to adult trees to yield realistic biomass damage estimates is still uncertain.' does not benefit from the suggested change of adding 'or not'.

**1.2 Additional note**

We updated the reference Oliver et al. 2017 (paper in discussion) to the final published paper Oliver et al. 2018.

[revised manuscript text omitted]